# Incorporating evolutionary and threat processes into crop wild relatives conservation

Crop wild relatives (CWR) intra- and interspecific diversity is essential for crop breeding and food security. However, intraspecific genetic diversity, which is central given the idiosyncratic threats to species in landscapes, is usually not considered in planning frameworks. Here, we introduce an approach to develop proxies of genetic differentiation to identify conservation areas, applying systematic conservation planning tools that produce hierarchical prioritizations of the landscape. It accounts for: (i) evolutionary processes, including historical and environmental drivers of genetic diversity, and (ii) threat processes, considering taxa-specific tolerance to human-modified habitats, and their extinction risk status. Our analyses can be used as inputs for developing national action plans for the conservation and use of CWR. Our results also inform public policy to mitigate threat processes to CWR (like crops living modified organisms or agriculture subsidies), and could advise future research (e.g. for potential germplasm collecting). Although we focus on Mesoamerican CWR within Mexico, our methodology offers opportunities to effectively guide conservation and monitoring strategies to safeguard the evolutionary resilience of any taxa, including in regions of complex evolutionary histories and mosaic landscapes.

Attaining food and nutrition security under global change poses major challenges for humanity. The importance of crop wild relatives (hereafter CWR) consists in their role as a substantial reservoir of genes that can help crops adapt to changing environmental conditions[1]. CWR are generally more genetically diverse than crops, and carry useful adaptations for a large set of biotic and abiotic stresses, like drought, pests, and diseases, as they have evolved to survive in diverse and often extreme environments. Planning for the conservation of genetic diversity of Mesoamerican wild relatives of some of the world's most important crops (e.g. maize, beans, chillie, pumpkins) is particularly urgent, as up to 35% of taxa recently assessed are threatened with extinction according to the International Union for Conservation of Nature (IUCN) Red List[2]. Many CWR are threatened by anthropogenic activities and habitat modification[2-4] as also are many other wild species in the mosaic landscapes that dominate in the Anthropocene. Thus, innovative conservation solutions are needed to retain the range-wide genetic diversity that can be at risk even in widespread species, which is crucial for maintaining evolutionary resilience[5,6].

Systematic conservation planning can aid the process of effectively locating and managing key areas to protect biodiversity by addressing the representation and persistence of diversity across all levels of biodiversity, as well as ecological and evolutionary processes. Nevertheless, most systematic conservation planning assessments focus on taxa representation rather than persistence, while at the same time, the representation of genetic diversity below the taxonomic species level (i.e. range-wide genetic diversity) remains understudied[5,7-9].

As in any wild species, CWR genetic diversity depends on demographic history, population structure, and natural selection[10,11], which changes according to the climatic and geologic history of a given area[12]. Most regions where crops were originally domesticated (centers of domestication)—often holding high diversity of CWR—are tropical or topographically heterogeneous[13], and therefore areas where long-term population persistence and historical isolation promote genetic differentiation in complex patterns[14,15]. To address the processes behind CWR genetic diversity, systematic conservation planning should consider both current and historical drivers of evolution.

e-mail: amastretta@conabio.gob.mx

Another challenge for planning approaches is to account for CWR interaction with agriculture. Most CWR are threatened by land use change linked to the expansion of intensive agriculture, although some thrive in managed and agricultural landscapes and are tolerated or fostered by farmers[2,16–19]. In addition, in some regions gene flow between crops and their CWR can occur, and has been part of the crop domestication process for thousands of years[18–21]. However, in modern agriculture, which is linked to the cultivation of highly genetically uniform commercial varieties and living modified organisms, gene flow can become a risk in the form of genetic assimilation (crop alleles replacing wild ones) and demographic swamping (wild populations shrinking due to hybridization), as well as causing other unintended consequences, like the evolution of new or more problematic weeds or invasive species[22,23]. Gene flow can also occur in the opposite direction, which some farmers perceive as a problem causing a lack of interest in CWR conservation[20,23,24].

Conservation planning for CWR has increased during the last few years[e.g.25–30], including the identification of priority sites for in situ

protection and ex situ collections, thus greatly contributing to CWR conservation globally[31]. Notwithstanding, most CWR planning apply the "minimum set cover problem" that detects the least number of sites to complement existing reserves, which may not fully represent the genetic diversity spectrum of any given species[6]. This is particularly important in centers of origin, diversity and domestication of crops, such as Mexico, where genetic diversity is driven by complex climatic, geologic and human histories[32], and where crops and their CWR co-exist under different forms of human management and have interacted over long periods of time[33–35].

To enable concentrated management efforts aimed at conserving crop wild relatives, here, we introduce a framework to assess conservation areas based on a hierarchical prioritization of the landscape in order to identify a portion of the country that maximizes the representation of genetic diversity in the absence of genomic information. The systematic conservation planning was designed to account for: (i) evolutionary processes, including historical and environmental drivers of genetic diversity, and (ii) threat processes,

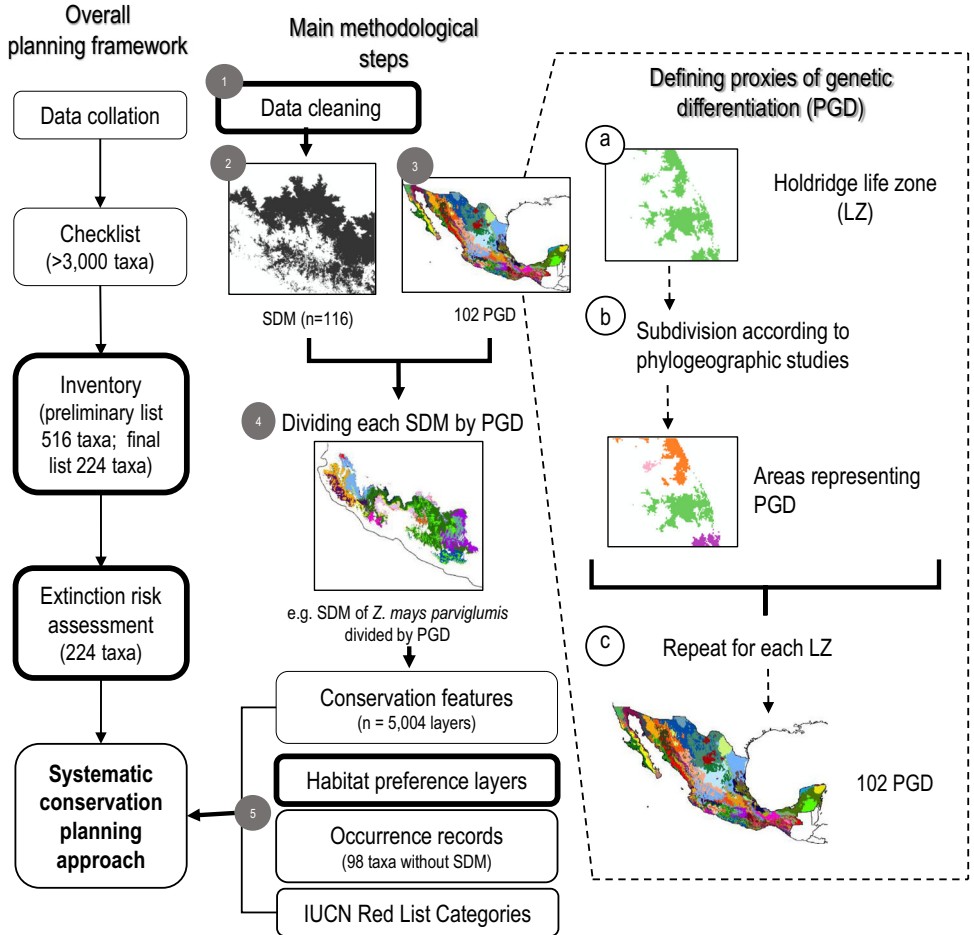

**Fig. 1 | General workflow of the project "Safeguarding Mesoamerican crop wild relatives".** Overall stages of the planning framework are shown at the left, and the main methodological steps to the right, including an approach to incorporate infraspecific genetic variation in a spatially explicit way. (1) Information, including data on occurrences, extinction risk, economic or social aspects, among others of taxa listed in the Mesoamerican CWR inventory was collated and cleaned. (2) Species distribution models (SDM) for 116 taxa were obtained. (3) In parallel, proxies of genetic differentiation (PGD) were generated for Mexico, integrating environmental and historical drivers of differentiation. For this, (a) Holdridge life zones (accounting for environmental variability) were (b) subdivided into areas that could be potentially isolated due to historical processes (using cartography on biogeographic provinces, edaphology, and watersheds) based on a literature

review on diverse taxa (Supplementary Data 8, 9, 10). The example shows a life zone that was subdivided in four 4 PGD. (c) Repeating this process for each life zone totaled 102 PGD among which it would be expected to find genetic differentiation among populations. (4) Each SDM was subsequently subdivided by PGD, so potentially differentiated populations could be recognized and included as different conservation features ("layers") in the spatial assessment. (5) The systematic conservation planning analysis to identify conservation areas for CWR in Mexico included: the SDM subdivided by PGD (5004 layers); taxon-specific habitat preferences considering land-use and land-cover information; occurrence records for taxa without SDM; and taxa weights according to their IUCN threat category. Boxes in bold indicate steps with experts' participation and assessments. [Spatial data is licensed under CC-BY 4.0; country boundaries according to Natural Earth.].

**Table 1 | Mesoamerican crop wild relatives selected for extinction risk assessment**

| Crop common name | CWR genus | No. of taxa | IUCN Red List Category | | | | | | | No. of taxa with potential species distribution model |
|---|---|---|---|---|---|---|---|---|---|---|
| | | | Critically endangered (CR) | Endangered (EN) | Vulnerable (VU) | Near threatened (NT) | Least concern (LC) | Data deficient (DD) | | |
| Chili pepper | Capsicum | 4 | | 1 | | | 3 | | | 4 |
| Squash | Cucurbita | 11 | | 1 | 2 | | 5 | 3 | | 10 |
| Cotton | Gossypium | 13 | 2 | 6 | 4 | | 1 | | | 6 |
| Avocado | Persea | 18 | | 7 | 2 | 1 | 5 | 3 | | 12 |
| Bean | Phaseolus | 55 | | 12 | 5 | | 36 | 2 | | 22 |
| Husk tomato | Physalis | 67 | 2 | 3 | 2 | 4 | 47 | 9 | | 30 |
| Potato | Solanum sect. Petota | 26 | | 5 | 1 | 2 | 16 | 2 | | 18 |
| Maize | Tripsacum | 11 | | 3 | 1 | 1 | 7 | | | 3 |
| Maize | Zea | 11 | 2 | 3 | 1 | | 5 | | | 9 |
| Vanilla | Vanilla | 8 | 1 | 6 | | | | 1 | | 2 |
| Total | 10 | 224 | 7 | 47 | 16 | 9 | 125 | 20 | | 116 |

considering taxa-specific tolerance to human-modified habitats, and their extinction risk status. To accomplish this, in the context of the project 'Safeguarding Mesoamerican crop wild relatives', a collaborative partnership between government agencies, local communities, universities, and non-governmental organizations from El Salvador, Guatemala, Honduras, Mexico, the UK, and IUCN, we applied a modified version of a planning framework for CWR conservation[25,26]. It included: (i) a CWR checklist, *i.e.* list of CWR taxa distributed in Mesoamerica, which were subset to reduce the number of taxa (Supplementary Data 1, 2), (ii) a CWR inventory, i.e., taxa selection and collation of ancillary data (Supplementary Data 3), (iii) a taxa extinction risk assessment (Supplementary Data 3), and (iv) systematic conservation planning analyses for supporting in situ and ex situ conservation. The first three steps are detailed in Goettsch et al.[2]. This study relies on introducing an approach in step (iv) to account for genetic differentiation in a spatially explicit way, through the use of proxies of genetic differentiation (Fig. 1). Specifically, we accounted for evolutionary processes by identifying geographic areas within the distribution of a taxon, considering environmental conditions and the isolation history of a given area, where interbreeding individuals can be expected to show some level of genetic differentiation from individuals of other areas. Hereafter, we call these areas 'proxies of genetic differentiation', and use them to represent the potential genetic differentiation within the distribution of a given taxon in a spatially explicit way, as they have a delimited non-overlapping geographical areas.

## Results

### Mesoamerican crop wild relatives distribution within Mexico

For this study, we used the inventory of 224 native taxa of Mesoamerican CWR (210 species and 14 subspecific taxa, including subspecies, and other infraspecific categories). They are related to nine crops: chili pepper (species of *Capsicum*), squash (species of *Cucurbita*), cotton (species of *Gossypium*), avocado (species of *Persea*), bean (species of *Phaseolus*), husk tomato (species of *Physalis*), potato (species of *Solanum* sect. *Petota*), maize (species of *Zea* and *Tripsacum*), and vanilla (species of *Vanilla*) (Table 1, Supplementary Data 3). Of these, seventy-four were previously identified as global priority CWR for Mexico[27]. According to the IUCN extinction risk assessment, seven of the 224 taxa are critically endangered, 47 endangered, 16 vulnerable and nine near threatened (Supplementary Note 1)[2].

Potential species distribution models (commonly and hereafter referred to as SDM) of Mesoamerican CWR with more than 20 occurrence records were used to overcome biases in sampling efforts (Supplementary Data 4, 5, 6); these were quality-checked by independent data and by integrating expert knowledge on each taxa[36]. Based on SDM, areas of high taxa richness were identified along the Trans-Mexican Volcanic Belt, and in the montane areas of Oaxaca and Chiapas in Southern Mexico (Fig. 2), which are known for harboring taxa with high genetic differentiation[32]. This spatial distribution pattern is also consistent with the global richness of CWR[37] and trends in other taxonomic groups showing higher species richness in heterogeneous and montane environments[38,39]. A global study using 1,076 CWR taxa highlighted montane areas of Mexico as showing high CWR richness with around 35 taxa[37]. Here, we used 116 SDM (i.e. ~10% of the taxa compared to the global analysis), which showed potentially more taxa per area than the global study (>50 taxa of CWR by square km; Fig. 2; >40 taxa of CWR using occurrence georeferenced data by 5 km², Supplementary Fig. 1, Supplementary Note 2). Considering the occurrence data and SDM of taxa within terrestrial protected areas, our results show that potentially there are seven protected areas with more than 20 taxa (Supplementary Data 7, Supplementary Note 3). Other CWR analysis focused on Mexico estimated up to 167 taxa in a 15 × 15 km grid square[27].

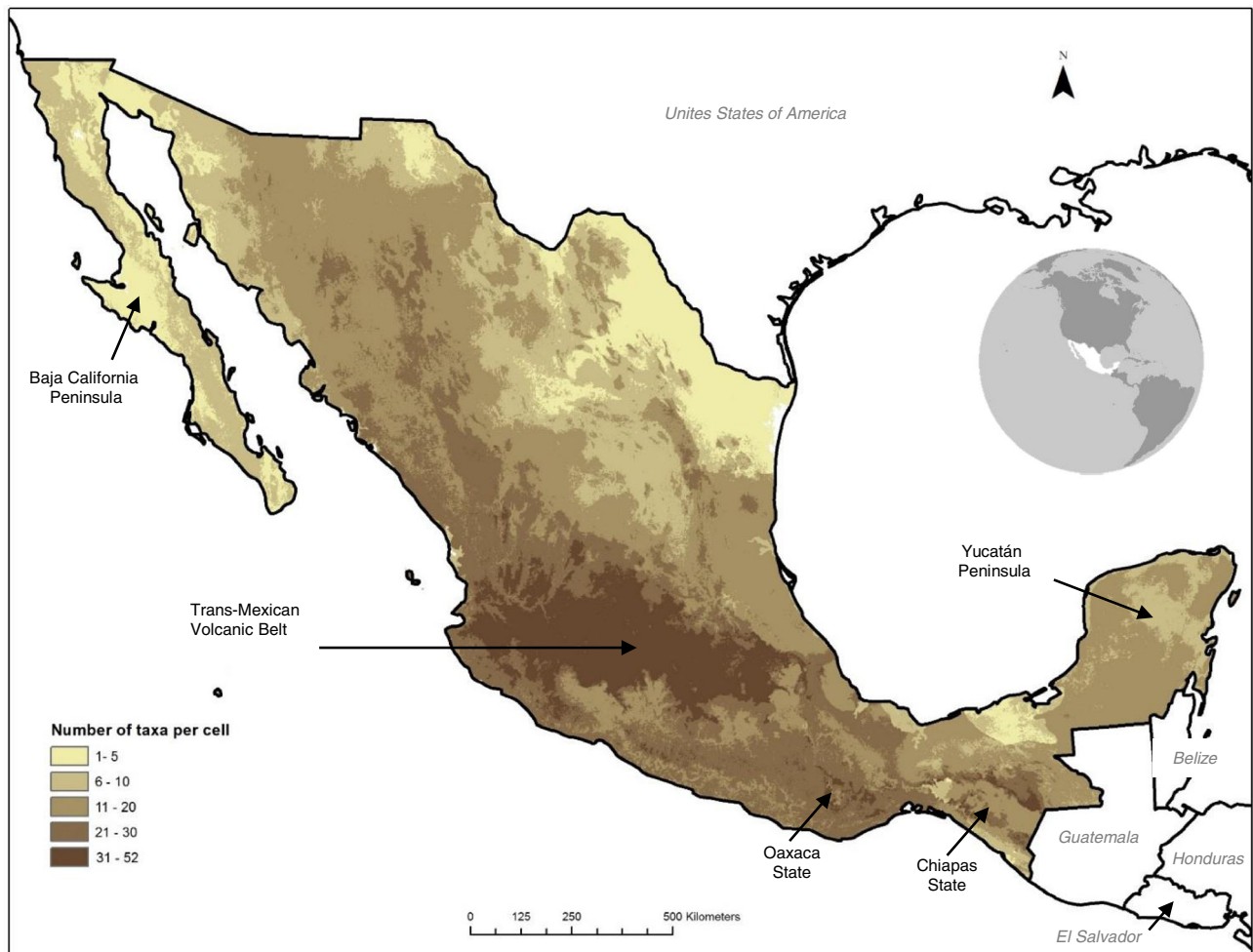

**Fig. 2 | Estimated Mesoamerican crop wild relatives richness in Mexico, based on 116 taxa with potential distribution models for which enough occurrence records were available.** Spatial resolution 1 km². Region names included are referenced in the main text. [Spatial data is licensed under CC-BY 4.0; the map was made with Natural Earth.].

## Overcoming lack of genetic data with spatial surrogates

Given that high-resolution genetic data for all taxa will not be available soon, we propose to use proxies of genetic differentiation as a spatially explicit surrogate of the potential genetic differentiation within the distribution of a given taxon, which can represent different populations across landscapes. Studies that considered genetic variation for spatial analysis of systematic conservation planning are extremely rare[40,41]. In the context of CWR conservation, Parra-Quijano et al.[42,43] introduced an ecogeographic land characterization that assumes that adaptive genetic features vary according to environmental variation. The approach was introduced as a 'proxy for genetic diversity', but may fall short to represent areas where population differentiation is expected due to historical processes, like range shifts during the Pleistocene glacial fluctuations or range's subdivision into naturally isolated populations, which are strong drivers of population structure in tropical and topographically complex areas[32]. Population structure can result in locally restricted alleles, both neutral and of adaptive value. For instance, an adaptive allele can emerge and be selected in a population, but never reach another population under similar environmental conditions, because there has been little or no gene flow among populations for long periods of time[44,45]. The passive accumulation of genomic divergence among populations can also lead to speciation, by processes other than natural selection alone[46]. Although isolation by distance is a common pattern, distance alone tends to not be a good surrogate for representing broad-scale genetic diversity,

because it has the potential to miss genetically distinct groups of populations[41]. Therefore, population structure should be accounted for when targeting to represent, conserve and monitor genetic variation[9]. Hence, to delimit proxies of genetic differentiation we focused on: (1) including not only environmental drivers of genetic differentiation, but also historical drivers that could lead to population differentiation and structure, and (2) defining proxies in an efficient, repeatable, and spatially explicit way that could be incorporated into systematic conservation planning analysis.

To account for environmental differences, we used Holdridge's life zone classification system, based on climatic driving factors (Supplementary Data 8, Supplementary Note 2), that can be estimated for any land area in the world. Then, to account for historical differentiation, we subdivided each life zone into areas that could be potentially isolated due to historical processes. For this, we first conducted a literature review on phylogeographic patterns for species of diverse taxonomic groups (Supplementary Data 9, 10). Then, we used these patterns to subdivide each life zone into several areas, each representing a different proxy of genetic differentiation. To translate this information into a spatial context, we used biogeographic regions, topographic or edaphic data to split the life zones into different subzones using the best fitting cartography representing the phylogeographic patterns (Supplementary Fig. 3). Since most life zones cover large territories, and complete phylogeographic congruence among different taxa is uncommon, we targeted phylogeographic patterns to

represent general trends that would likely hold across species (Supplementary Fig. 4). This allowed us to delimit areas known to have differentiated populations, despite showing similar environmental conditions, e.g. West/East of the Tehuantepec Isthmus[47] or among the main Mexican mountain ranges[32]. Thus, we obtained 102 proxies of genetic differentiation for Mexico; each proxy has a particular distribution area, unique in shape, size, and location that does not overlap with other proxies (Supplementary Fig. 5). They can further be used to inform conservation assessments and actions in the country.

To test how well our method worked for identifying proxies of genetic differentiation, we leveraged available empirical data[48] from the teosinte *Zea mays* subsp. *parviglumis* (a wild relative of maize), which was excluded from the literature review to avoid having its phylogeographic patterns influence the estimation of proxies of genetic diversity for this taxon. This approach allowed us to mimic the scenario presented by the many species without any existing genomic resources. According to a population analysis with >30,000 single nucleotide polymorphisms (SNPs) data, *Z. mays* subsp. *parviglumis* is structured in 13 genetic clusters which includes a longitudinal gradient as well as some considerably differentiated genetic clusters (Fig. 3a, c). Normally, systematic conservation planning would use SDM without differentiating genetic clusters within the taxon distribution range, thus stochastically representing genetic groups based on their geographic extent (Fig. 3d). However, our approach allowed us to increase the representation of areas with likely differentiated populations (Fig. 3d, Supplementary Fig. 6). For example, the Zonation scenario using only SDM favors the representation of populations that fell in the proxies 37, 5, 41 and 36 (Fig. 3d) and poorly represents populations that fell in proxies 8 and 40 (dark blue admixture group, Fig. 3a, b). Alternatively, the Zonation scenario dividing SDM by proxies of genetic differentiation ("SDM*PGD", as described below), increases the representation of all proxies, so that no admixture group is poorly represented (i.e. populations PGDs 48, 10, 11, 8, and 41 are better represented relative to the SDM scenario, Fig. 3d). Therefore, although there is no complete coincidence between the proxies of genetic differentiation and the empirical genetic data (i.e. each genetic clustering matching only one proxy in Fig. 3a), using proxies maximized the representation of genetic differentiation in the spatial analysis, in contrast with considering all the taxon distribution as a single unit (Fig. 3d). In other words, although the proxies of genetic differentiation are not perfect, they are better at representing the genetic variation in the Zonation output than if using SDM alone.

Proxies of genetic differentiation can be assessed for any taxa without genetic data, which is particularly important if we aim to secure genetic variation within all taxa[42]. Even if no literature on phylogeography is available, it might still be possible to assume genetic variation based on biogeographical, climatic and geological conditions that have shaped biodiversity and diversification patterns[26]. For example, countries like El Salvador used national or global datasets on mountain areas and tectonic plates—as they drive geology and biogeographical patterns—in order to assess genetic proxies (see conservation areas on the project report including this country[49]), but more empirical data is needed to test this.

Of course, as any given model or surrogate, the proxies of genetic variation are a simplified version of the reality; thus, the main caveat of our methodology is that the spatial expression of the proxies of genetic differentiation would never be as accurate as actual genetic studies. However, as genomic resources become available for more species, the proxies of genetic differentiation could be further corroborated and fine-tuned. For example, with minor changes to the proxies 5, 10, and 36 (Fig. 3), the empirical genetic structure of *Zea mays* subsp. *parviglumis* would be better reflected. Thus, regardless of the quality of the proxies, our analyses are also a proof of concept of how genetic differentiation can be incorporated in a spatially explicit way into systematic conservation planning.

### Importance of using proxies of genetic differentiation in systematic conservation planning

Based on the support from empirical genetic data, we assumed that the delimitation of all taxa ranges given the proxies of genetic differentiation best reflected infraspecific variation (i.e. populations holding differentiated genetic diversity). We quantitatively tested different alternatives to incorporate proxies of genetic differentiation into the conservation analysis (Supplementary Fig. 7) and compared it against results excluding them, or using only SDM or Holdridge's life zones (Fig. 4). The approach that maximized the representation of intraspecific diversity as given by the measure of the proportion of proxies of genetic diversity areas averaged for each taxon, used a dataset where each SDM was subdivided by the genetic proxies ("SDM*PGD" scenario). While a somewhat redundant measure, it allowed us to determine that any analysis shortcut would be significantly suboptimal, as shown by performance curves (Fig. 4). The combination of SDM and proxies resulted in 5004 input layers or conservation features; each conservation feature represented a part of a taxon range occurring in a proxy of genetic differentiation (theoretically, a taxon could be represented by max. 102 individual layers if all proxies of Mexico would cover a SDM). Based on the "SDM*PGD" scenario, on average 41% of each taxon range and 76% of the area of each proxy of genetic differentiation within each taxon range were represented when evaluating 20% of the country (Supplementary Fig. 8). Threatened taxa were the best represented both in terms of their range and proxies of genetic differentiation within them. The less represented taxa tended to be widely distributed, in which only 25% of their range was included in the solution for 20% of Mexico, but, on average, more than 50% of the area of each proxy of genetic differentiation is represented within these taxa.

Other scenarios also captured the potential genetic variation inferred through proxies, but were less efficient (Fig. 4). When considering the potential distribution models and proxies of genetic differentiation independently ("SDM + PGD" scenario: a taxon was represented by one SDM, and each proxy of genetic differentiation were included as an individual conservation feature), or using proxies of genetic differentiation as administrative units ("SDM and PGD as ADMU" scenario), on average, 37% and 48% of the range of each taxon was represented in 20% of the country, respectively, but the average area of each proxy of genetic differentiation within each taxon range was smaller compared to other scenarios ("SDM" scenario: 57%; "SDM and PGD as ADMU" scenario: 66%; "SDM*PGD" scenario: 76%). Performing the analysis only with potential distribution models ("SDM" scenario), resulted in the highest proportion of area of taxa ranges (on average 48%), but the poorest representation of proxies of genetic differentiation within the area of each taxon (on average 54%). Therefore, using only SDM as conservation features was less efficient in representing genetic diversity than alternative scenarios. This suggests that performing conservation analyses without explicitly considering an indicator for genetic differentiation is likely to lead to poor coverage of range-wide genetic diversity in proposals of conservation networks.

### Conservation areas for Mesoamerican CWR in Mexico

In order to identify areas for safeguarding Mesoamerican CWR in Mexico we incorporated in the final systematic conservation planning analysis: (a) CWR potential distribution models subdivided by proxies of genetic differentiation (i.e. areas that potentially represent genetically distinctive populations, assuming that they have adapted to particular climatic conditions or have been split by historical processes; see "SDM*PGD" scenario, Fig. 4), so potential populations of species (and infraspecific levels) could be recognized as different features in the spatial assessment; (b) the IUCN threat categories to weight taxa (giving higher values to taxa at higher risk of extinction; Supplementary Data 3); (c) occurrence records for taxa without potential distribution model, to ensure their representation in the

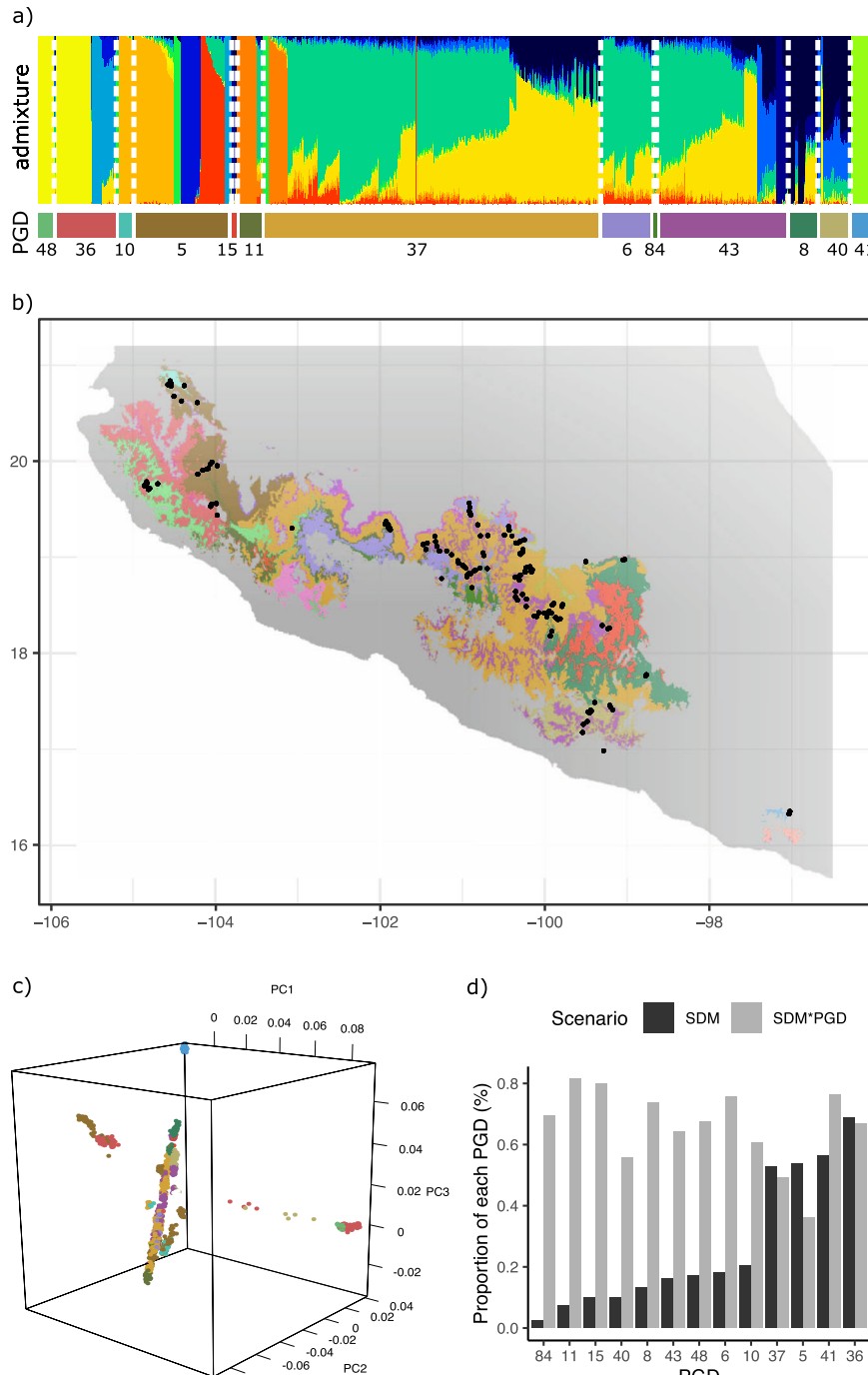

**Fig. 3 | Genetic diversity of *Zea mays* subsp. *parviglumis*, a maize CWR, represented in the proxies of genetic differentiation (PGD).** (**a**) Admixture plot assuming *K* = 13 genetic clusters, using ca. 30,000 SNPs (data from Rivera-Rodrígue z[48]). Each bar represents the proportion of different genetic clusters (colors) conforming an individual. White dashed lines separate the proxies of genetic differentiation (numbered colored bars below, matching colors in the map of (**b**), where the samples fell. Colored bars only include proxies with sampling points used in the genetic analysis. (**b**) Potential distribution model of *Z. mays* subsp. *parviglumis* subdivided by proxies of genetic differentiation (background colors), overlaying the geographic location of individuals sampled for genetic analyses (black dots). (**c**) Score plot of a principal component analysis performed with the genetic data. The first three components are projected. Each point represents an individual colored by the proxy of genetic differentiation where it fell according to (**b**). (**d**) Proportion of the area of each proxy of genetic differentiation represented in the Zonation

solution of the preliminary analyses given two different SCP scenarios (only considering SDM, or combining SDM*PGD), considering 20% of Mexico's terrestrial area. See Supplementary Fig. 7 for how the mean proportions were estimated. Most of the spatial extent of the SDM scenario (traditional approach for SCP) covered only a few genetic clusters, including the Western (PGD 36 and 5), Center-Eastern (PGD 37) and Eastern (PGD 41) distribution of the taxon. Contrarily, the SDM*PGD scenario (approach proposed here) increased the representation of other areas with populations likely differentiated, like the clusters represented in PGDs 48, 10, 11, 8, and 41. Thus, although the proxies of genetic differentiation are not a perfect match to the empirical population differentiation within this taxon, they maximized the representation of genetic differentiation in the spatial analysis as shown in Fig. 5, in contrast with considering all the taxon distribution as a single unit. Source data are provided as a Source Data file. [Spatial data is licensed under CC-BY 4.0; country boundary according to Natural Earth.].

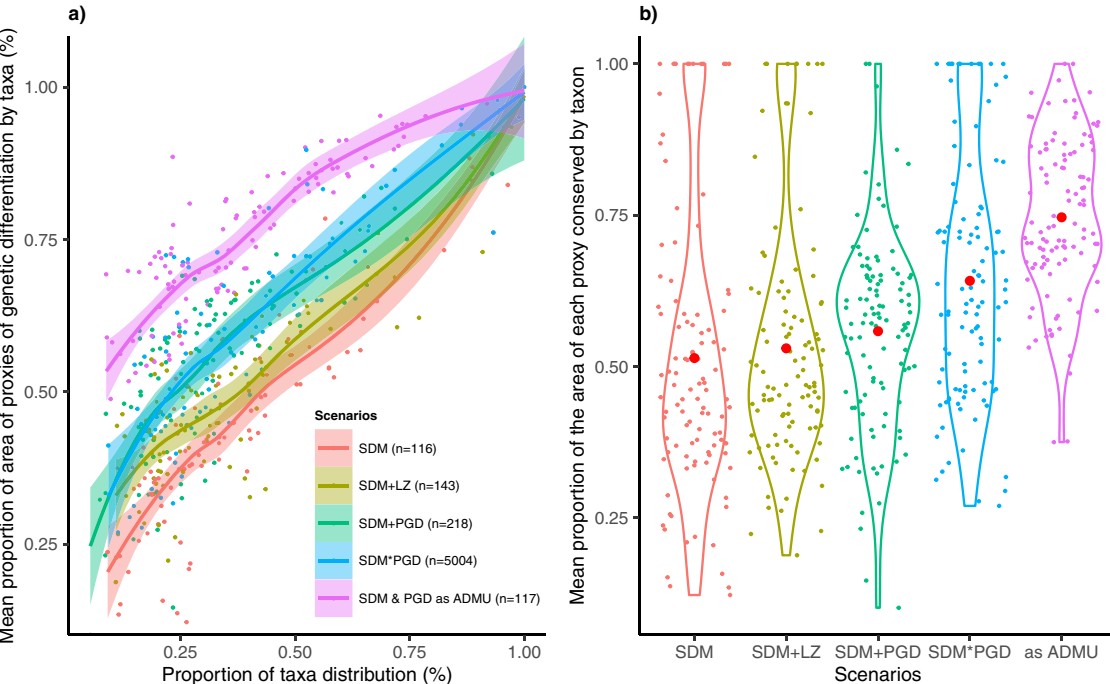

**Fig. 4 | Performance of five systematic conservation planning scenarios to represent conservation features of Mesoamerican crop wild relatives, considering 20% of Mexico's terrestrial area.** (a) Representation curves of the proportion of taxa distributions and mean proportion of proxies of genetic differentiation (PGD) areas within them. Smoothing method was locally estimated scatterplot smoothing. Confidence intervals (0.95 confidence level) around the fitted line are shown. (b) violin plots showing the mean proportion of the area of each proxy represented within each taxon (red dots correspond to the median, colored points correspond to each taxon). Scenarios: (i) SDM, considered 116 taxa with potential species distribution models; (ii) SDM + LZ, considered 116 taxa with potential distribution models and 27 life zones; (iii) SDM + PGD, considered 116 taxa with potential distribution models and 102 proxies of genetic differentiation; (iv) SDM*PGD, considered 5004 conservation features, that resulted by subdividing each of the 116 potential distribution model by 102 proxies of genetic differentiation (as some combinations produced empty outputs given the extension of SDM that do not cover all of Mexico, we only used the layers with taxa information as conservation features); (v) SDM and PGD as administrative units (ADMU), considered 116 taxa with potential distribution models as conservation features and the proxies of genetic differentiation as one single layer considering 102 units of analysis (we used the ADMU function that allowed to consider each proxy as an independent planning unit to guarantee the representation of taxa). Source data are provided as a Source Data file.

solution (Supplementary Data 3); and (d) taxon-specific habitat preferences to identify suitable locations considering land-use and land-cover information, including different types of agricultural systems (Supplementary Data 11, Supplementary Fig. 9). To incorporate all these data into the analysis, we used the Zonation software[50,51], that establishes a hierarchical prioritization of the landscape and optimizes the representation of taxa or other conservation features in a given area.

The Convention on Biological Diversity Aichi Target 11 encouraged parties to protect at least 17% of terrestrial regions[52]. Therefore, we established an area threshold of 20% of the hierarchical map (Fig. 5a), which were proposed as areas for CWR in situ conservation (Fig. 5b). The 20% area was also based on the performance curves to efficiently represent taxa ranges delimited by proxies of genetic differentiation (Fig. 6); we used the data to evaluate the results. The identified conservation areas are located in the temperate mountain areas of the Trans-Mexican Volcanic Belt, characterized by high species richness and endemism; in the region from central Veracruz state to Chiapas, crossing Puebla and including large areas of Oaxaca, the Tehuacán-Cuicatlán valley and the Chimalapas region, corresponding to areas of highly heterogeneous environments; along the northern coastline of the Michoacán state in Central Mexico, and in the cloud forests and rainforests of southern Mexico, representing habitats for range-restricted species such as *Vanilla odorata*; and in the arid and semi-arid areas of the states of Sonora and Baja California, where *Gossypium* is common (Fig. 5a, b). Almost half of these important areas for conservation were located within areas where indigenous communities live (Fig. 5b). This is not surprising because of the biocultural

relationship between the possible descendants of the people who putatively started the domestication of crops, and the biological diversity available in the areas where they settled. Since indigenous communities continue to traditionally manage plant diversity, including CWR, these areas are of particular relevance for the maintenance of evolution under domestication[16–19]. Also, 11% were located within federal protected areas (Fig. 5b), and one third of areas voluntarily destined for conservation are covered in the selected 20% area. Protected areas represent great opportunities for active conservation and implementation of management and monitoring of CWR[39], but because areas voluntarily destined for conservation are a biocultural approach for sustainable management, they in particular offer an opportunity to further support CWR conservation. For all kinds of protected areas, a needed step to promote CWR conservation in management plans is to generate comprehensive inventories of CWR occurring within them[2,53], along with other conservation approaches[2].

Performance curves showed the effectiveness of the spatial solution at representing CWR potential distribution ranges and likely its genetic diversity in a top fraction of land (Fig. 6, Supplementary Fig. 10). For instance, in 20% of the country, on average, 50% of the area of each taxon within each proxy of genetic differentiation was represented. Representation values of taxa grouped by threat categories differed due to the conservation weights established according to their extinction risk. Of note, it was impossible to represent 100% of the taxa ranges, and consequently their proxies of genetic differentiation, given that taxon-specific habitat preferences were included in the analyses. It was revealed that for many taxa a considerable amount of their habitat has already been lost, degraded, and fragmented within their potential

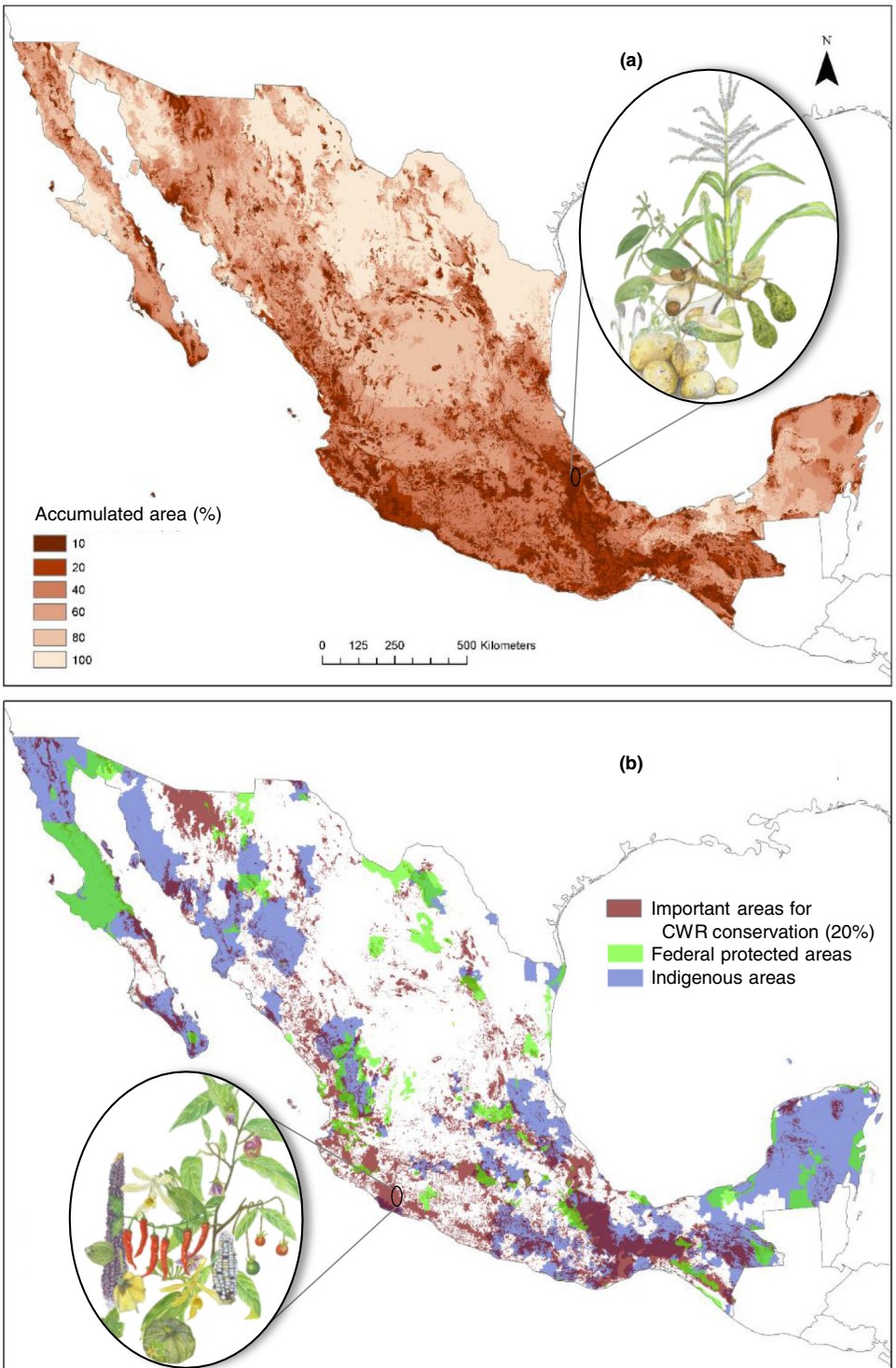

**Fig. 5 | Results of the systematic conservation planning process for Mesoamerican crop wild relatives in Mexico, based on the scenario considering potential distribution models subdivided by proxies of genetic differentiation, SDM*PGD (see Fig. 4, scenario iv), and including occurrence records for taxa without SDM, IUCN threat categories, and specific habitat preferences (see text for details).** (a) Hierarchical landscape priority rank map, where the 10% most valuable fraction is within the most valuable 20% fraction; the most valuable 20% is within the most valuable 30%; thus expressed as accumulated area. (b) Conservation area proposal considering 20% of Mexico's terrestrial area to maximize the representation of taxa and proxies of genetic differentiation (see Fig. 6). Federal protected areas and areas where indigenous communities live are displayed to show the coincidence with these areas, although the criteria were not considered in the spatial analysis. [Spatial data is licensed under CC-BY 4.0; country boundaries according to Natural Earth. Images of the following taxa were included to show its subsistence in a given area: (**a**) *Phaseolus maculatus, Solanum tuberosum, Persea americana, Persea schiedeana, Tripsacum pilosum, Vainilla inodora, Zea mays* subsp. *mays*. (**b**) *Capsicum annuum* var. *annuum, Capsicum lanceolatum, Cucurbita pepo* subsp. *pepo, Phaseolus maculatus, Physalis philadelphica, Vanilla planifolia, Vanilla pompona*, and *Zea mays* subsp. *mays*. Illustrations by Adriana Iwasaki Otake and Héctor Tobón y Hernández, licensed under CC BY-NC-ND 4.0.].

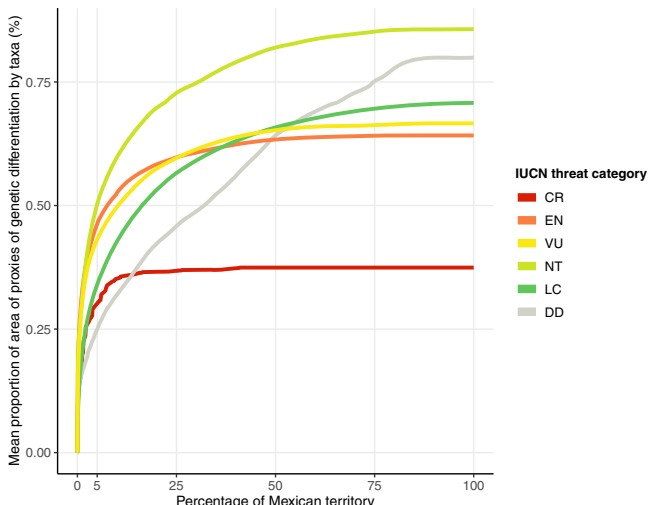

**Fig. 6 | Performance curves quantifying the representation of proxies of genetic differentiation within the distribution of crop wild relatives in Mexico, based on the hierarchical landscape priority rank map (Fig. 5a) which is based on the "SDM*PGD" scenario, considering potential distribution models subdivided by proxies of genetic differentiation (see Fig. 4), and including occurrence records for taxa without SDM, IUCN threat categories, as well as specific habitat preferences (see text for details).** The information on habitat of CWR included in the analysis revealed that a substantial amount of habitat had already been converted to other land uses, thus it was no longer possible to completely represent taxa distribution ranges and the associated intra-specific variation. Data is grouped by IUCN Red List Category of taxa (CR critically endangered, EN endangered, VU vulnerable, NT near threatened, LC least concern, DD data deficient). Performance was evaluated for 116 taxa with potential species distribution model. Source data is provided as a Source Data file.

range (Fig. 6). This is particularly critical for the most threatened taxa, as on average more than 40% of their potential distribution ranges may no longer have suitable conditions. It indicates the urgent need for both conservation and restoration actions to maintain and recover natural vegetation, but might also imply that some genetically distinct populations could have already been lost. Local extinction of populations has been suggested as an important indicator of loss of genetic diversity[54], and it is expected to occur at an alarming rate[55].

Although CWR taxa might be represented within a relatively small area (i.e. in <2% of Mexico), the conservation of ecological and evolutionary processes shaping biodiversity at all levels (genes, populations, species, ecosystems) cannot be secured in a small fraction of the territory and with few individuals (Fig. 6). To represent at least 50% of all conservation features, 80% of Mexico's terrestrial surface must be sustainably managed. Effective population size (normally equivalent to the number of breeding individuals contributing with their genetic variation to the next generation) within each differentiated population needs to be large enough to conserve genetic diversity[54,56,57]. Conserving CWR differentiated populations with large effective population sizes is particularly important for a country like Mexico, where the processes related to the coevolutionary dynamics of plant domestication continues to occur today in the hands of millions of smallholder farmers, and where crops also are expected to have large effective population sizes[56]. Therefore, the hierarchical priority rank map (Fig. 5a) and selected conservation areas (Fig. 5b) offer a guide for implementing tailored regional and local conservation and sustainable landscape management measures in Mexico at large spatial areas, e.g. landscapes, ecoregions, or basins to support biodiversity, including agroecosystems, and the services and benefits they provide[56]. Directed conservation actions and sustainable resource management at landscape level across the country, acknowledging the importance of CWR populations, the connectivity among them, and their interactions with

domesticated species, would support critical ecological and evolutionary processes, but see Contreras-Toledo et al.[27] for a representation approach at pixel level.

Since taxa have considerably distinct habitat preferences and life histories (Supplementary Data 11), they can be affected differently by land use change and agricultural practices:[8] e.g. *Persea* are trees distributed in well-preserved vegetation, such as cloud forests, while *Gossypium* are bushes requiring a certain degree of disturbance[43], and *Capsicum* and *Physalis* are managed in the wild or tolerated within differentiated human management and traditional agricultural systems, mainly within indigenous territories[17,58]. To identify conservation priorities for different land use preferences, we ran three scenarios, including: (a) all taxa, (b) taxa exclusively distributed in well-preserved vegetation, and (c) taxa that can be associated with different habitats and land uses (e.g. natural vegetation, agriculture and urban areas). Results showed different spatial solutions (Supplementary Fig. 11) and performance curves (Supplementary Fig. 12; Supplementary Note 4). although the general patterns in the largest aggregates areas are the same (i.e. areas where the three scenarios coincide; Supplementary Fig. 13). Also, we found that almost half of the area is located inside areas where indigenous communities live, and 11% of the selected area is covered by federal protected areas (Supplementary Fig. 14, Supplementary Data 12), although the layers were not considered as additional criteria to guide the solution. The results expose the habitat preferences of taxa targeted in each scenario (Supplementary Fig. 15). The scenario (b), considering taxa exclusively distributed in natural vegetation, showed a higher proportion of area in primary and secondary vegetation, while the scenario (c) based on taxa associated with different habitats showed a higher proportion of area in rainfed and moisture agriculture. Explicitly accounting for the effect of land use on conservation of CWR as done here can allow promoting synergic planning and actions among different sectors, especially between environment and agriculture[2]. Further analyses could also consider future land use and climate change scenarios to assess conservation priorities in the long term.

While our analysis has focused on in situ conservation, it could also be useful to address the challenges of ex situ conservation. Namely, the spatial results may indicate sampling areas based on either taxon rich sites or where range restricted and PGD-restricted populations are distributed (Fig. 5). Other areas of close attention for ex situ conservation actions should be representation gaps in poorly explored areas, considering the representation of genetic diversity.

In summary, our approach identified conservation areas of high CWR taxa richness and uniqueness, maximizing the representation of genetic diversity in a spatially explicit way by accounting for historical and environmental drivers of genetic differentiation. The major limitation of our study is the lack of high resolution genetic data to corroborate if the proxies of genetic differentiation are reliable across different taxa, and for fine-tuning the approach. However, given the rate at which biodiversity is declining, it is better to include an inaccurate representation of genetic diversity within taxa, than to perform conservation assessments without explicitly accounting for it at all. Our approach might be challenging in terms of preparing and handling large datasets; running analysis with thousands of input layers needed to be done in a computing cluster. Still, a major benefit of the Zonation software is its ability to incorporate different sources of data (i.e. SDM, occurrence records, proxies of genetic differentiation, land use and cover maps, and threat categories), and to link each biodiversity feature to a certain condition group, e.g. habitat preference. One of the software outputs is a hierarchical representation of the landscape that can inform proactive and reactive conservation measures (Supplementary Fig. 11), as well as connectivity when considering a conservation area proposal of 20% of the country (the maps of all three scenarios showed a clear aggregation of conservation areas, even though connectivity was not targeted in the analysis; Supplementary

Figs. 14, 15). Also, we followed the maximal coverage approach where it is not necessary to define specific area representation targets for each biodiversity feature; sites are ranked according to occurrence levels, conservation weight of biodiversity features, and other considerations. A general advantage of our approach is that it allows incorporating key information to enhance biodiversity conservation by addressing genetic diversity as well as environmental and evolutionary processes at the landscape level. Our framework can be used for the establishment of conservation areas, and also to promote sustainable management across landscapes. Thus, conservation and development goals can be tackled simultaneously in order to achieve long-term sustainability.

## Discussion

Conservation of genetic diversity of crops and their wild relatives is key to tackling the global environmental and social crisis, as communicated in the UN Sustainable Development Goal 2.5 (https://sdgs.un.org/goals) and Aichi Target 13 (https://www.cbd.int/sp/targets/) of the Convention on Biological Diversity. To move this field forward, we propose an approach to assess and incorporate indicators of genetic variation to identify areas of high conservation value for Mesoamerican CWR in Mexico, which is needed for strategic planning and decision-making at local, national, and regional scales. We focused on CWR due to their importance for food security, human well-being, the adaptation of crops to changing environments[1], as well as their relevance for contributing towards achieving international commitments[52]. However, as recent suggestions to the Convention on Biological Diversity post-2020 strategy highlight, we also need to conserve and monitor genetic diversity beyond domesticated species and their wild relatives[54]. A first key step to do so is to spatially delimit genetically differentiated populations for which genetic data is not available[54]. Our proxies of genetic differentiation methodology could be applied to any taxa to achieve this goal, and thus represents a contribution to incorporate genetic diversity monitoring and conservation into national and regional biodiversity management strategies, which is needed to enhance species evolutionary resilience in the face of climate change and other threats[5]. As better phylogeographic meta-analysis and genetic data become more available, proxies of genetic differentiation could be fine-tuned or delimited at higher resolutions.

Our proposed systematic conservation planning analysis not only allows maximizing the representation of genetic diversity through explicitly considering PGD, as well as the representation of threatened and range limited taxa, it also allows accounting for taxa-specific tolerance to human-modified habitats. The hierarchical representation of the landscape offers a broader perspective not only to identify where area-based conservation measures are mostly required, but also to implement sustainable development policies in agricultural landscapes that strengthen rural communities and economies[59]. In order to do this, local communities and their visions, objectives and needs should be incorporated in the conservation programs.

Our results support the development of National Strategic Action Plans for the conservation and use of CWR in Mexico, inform public policy regarding crops living modified organisms[33] and agriculture subsidies[34] to mitigate threat processes to CWR[2], as well as indicating future research needs, e.g., for potential germplasm exploration and collecting. Also, incorporating planning outputs to the design of cross-sectorial policies can allow moving in situ conservation of CWR beyond protected areas. This is of particular relevance for regions that are centers of origin, domestication and diversification of crops, and where sustainably managed landscapes can not only contribute to halting biodiversity loss[60,61], but also contribute to the provision of evosystem services (i.e. uses or services to humans that are produced from evolutionary process)[56,62].

## Methods

We applied a modified version of a planning framework for CWR conservation[25,26] which has been used by numerous countries of Europe[e.g.29,63,64], America[e.g.65], Africa[30] and Asia[66,67]. We addressed the following main steps of the toolkit (see Spanish version[49]): (i) CWR checklist, i.e., creating a list of CWR taxa distributed in an area (Supplementary Data 1), (ii) CWR inventory, i.e., taxa selection and collation of ancillary data, including taxonomic data (Supplementary Data 2), (iii) taxa extinction risk assessment (Table 1, Supplementary Data 3), and (iv) a systematic conservation planning assessment, i.e., spatial analyses to assess conservation areas (Fig. 1). We only provide a brief description of steps i-iii, as these are thoroughly described in Goettsch et al.[2]. Here, we focus on the systematic conservation planning assessment, introducing an approach in order to identify conservation areas for CWR that account for genetic differentiation in a spatially explicit way, through the use of proxies of genetic differentiation (Fig. 1).

During the process -framed under the project "Safeguarding Mesoamerican crop wild relatives" (https://www.darwininitiative.org.uk/project/23007/)- more than 100 experts from academic, governmental, and non-governmental organizations from El Salvador, Guatemala, Honduras, Mexico, the UK, and IUCN participated in six workshops, shared data, and provided fundamental knowledge and feedback at each project stage to ensure accurate, reliable and robust information for next steps. The checklist, inventory and risk assessment were collaboratively developed between partners of El Salvador, Guatemala, and Mexico (hereafter, Mesoamerica; Goettsch et al.[2]). The spatial analysis to identify areas for in situ and ex situ conservation of CWR was done independently by each country.

To assess conservation areas of CWR in Mexico, we developed proxies of genetic differentiation that account for evolutionary processes by including historical and environmental drivers of genetic diversity (see the Methods section 'Proxies of genetic differentiation'). In addition, we used criteria such as information on taxon-specific tolerance to human-modified habitats and IUCN extinction risk category. We applied a systematic conservation planning approach and performed spatial analysis using the software Zonation[50]. We compared different scenarios to represent genetic diversity of CWR based on potential species distribution models (SDM) and proxies of genetic differentiation.

### Study area

Mesoamerica is a cultural region encompassing the territories of Belize, Guatemala, El Salvador, the southern part of Mexico and parts of Honduras, Nicaragua and Costa Rica[see 2]. In this study, we also included the dry areas of northern Mexico that are part of Aridamerica[68] and the Nearctic biogeographic realm[69] to account for the full extent of the geographic range of many taxa included in the extinction risk assessment[2].

For the assessment of conservation areas, we focused on Mexico, which is one of the most biodiverse countries in the world[70]. The Mexican territory covers 80% of the landscapes of the region called Mesoamerica. Its high biological diversity is attributed to its geographic, topographic, climatic, geological and cultural characteristics, which, among other factors, shaped the distribution of an extraordinary variety of ecosystems and species with high levels of endemism and species turnover among different regions[32,71–73]. In particular, the high genetic variation within populations of landraces and CWR is the result of past and ongoing sociocultural processes occurring in a wide range of distinct environmental conditions[74,75].

### (i) CWR checklist and (ii) CWR inventory

The compiled CWR checklist included ~3000 species and subspecies of 92 genera and 45 families of plants that belong to the same genus of a crop cultivated in Mesoamerica, or wild plant collected for food or other uses in the region (Supplementary Data 1).

The first set of criteria were established in preparation for the first stakeholder workshop. The following criteria were applied at the genus level to compile the CWR inventory: (1) occurrence of wild relatives of cultivated plants or crops that were domesticated in Mesoamerica; (2) existence of research groups working on taxa that could support the extinction risk assessment; and (3) relation to a crop of economic and nutritional importance at local, national and regional levels, or cultivars known to require genetic improvement.

To narrow the list for the inventory and extinction risk assessment, similar criteria were agreed upon in the same workshop and applied at the species level: (1) native distribution in Mesoamerica, incl. Aridamerica; (2) related to a crop of economic or social importance based on production and nutritional value; (3) related to a taxon for which Mesoamerica is the center of origin or domestication; (4) constitutes part of the primary or secondary gene pool, and in some cases the tertiary gene pool[76]. The primary gene pool consists of wild plants of the same species as the crop and thus their mating produces strong fertile progeny. The secondary gene pool is composed of wild relatives distinct from cultivated species but closely related as to produce some fertile offspring (same taxonomic series or section in the absence of crossing and genetic diversity information, see the 'taxon group' concept proposed by Maxted and collaborators[77], Supplementary Note 5). The tertiary gene pool (same subgenus in the taxon group concept) corresponds to CWR that are more distant relatives to the taxa of the primary gene pool, but can have important adaptive traits which can be used with specific breeding techniques. This provided a preliminary list of 514 CWR taxa related to avocado, cotton, amaranth, cocoa, squash, sweet potato, chayote, chili pepper, cempasuchil, bean, sunflower, maize, papaya, potato, vanilla, and yuca (Supplementary Data 2).

The list had to be further reduced due to time and funding restrictions to include those genera which when added together would include no more than 250 taxa, and that the taxonomic groups could be comprehensively assessed and their taxa evaluated throughout their entire range. Thus, not all species in the group necessarily met the criteria previously mentioned. See the final Mesoamerican CWR inventory in Supplementary Data 3; see summary in Table 1.

### (iii) Taxa extinction risk assessment

Full methodological details and results of this section are described in Goettsch et al.[2]. Summarizing, during the process 224 taxa were evaluated according to the International Union for Conservation of Nature, IUCN, Red List Categories and Criteria[78]. The IUCN Red List is a critical indicator to identify species most vulnerable to extinction considering a set of criteria, i.e., species' population trends, size, structure, and geographic ranges. A Red List workshop with the participation of 25 experts from different project partner institutions and IUCN specialists was organized to assess the extinction risk of taxa. The threat analysis included not only species, but subspecies and subpopulations (i.e. races) for some groups (Supplementary Data 3, see summary in Table 1).

### (iv) Systematic conservation planning assessment

To undertake the following spatial analyses we focused on the dataset of 224 CWR described above, which is representative of the CWR of the main crops of Mesoamerica (10 genera, Table 1).

### Species distribution modeling

To compile occurrence records, hundreds of data sources were consulted, including published and personal databases of the project participants[e.g.79–82], the Agrobiodiversity Atlas of Guatemala (https://www.ars.usda.gov/northeast-area/beltsville-md-barc/beltsville-agricultural-research-center/national-germplasm-resources-laboratory/docs/atlas-of-guatemalan-crop-wild-relatives), the Global Biodiversity Information Facility (GBIF, https://www.gbif.org/), and Mexico's Biodiversity Information System (SNIB, http://snib.mx/).

To generate potential species distribution models (SDM), we used more than 13,000 occurrence records (Supplementary Data 4), that were standardized and curated by experts to generate the range maps of taxa as part of the extinction risk assessment, which were published in IUCN Red List (https://www.iucn.org/news/species/202109/threats-crop-wild-relatives-compromising-food-security-and-livelihoods). Spatial resolution of the SDM was 1 km². SDM were obtained for taxa with more than 20 unique occurrence data in a 1 km² grid covering the study extent to reduce uncertainty when using smaller sample sizes[83]. We used 19 bioclimatic variables and other climatic variables, such as annual potential evapotranspiration, aridity index, annual radiation, slope, and altitude[84–86]. Climate data represents annual and seasonal patterns of climate between 1950 and 2000. Also, we used a variable that described the percentage of bare soil and cultivated areas[87]. Collinearity between variables was assessed with the 'corselect' function of the package fuzzySim version 1.0[88], using a value of 0.8 and the variance inflation factors as criteria to exclude highly correlated variables.

We used MaxEnt version 3.3.1, a machine-learning algorithm that uses the maximum entropy principle to identify a target probability distribution, subject to a set of constraints related to the occurrence records and environmental data[89,90]. Model calibration area for each taxon included those ecoregions where the taxon has been recorded; we used the terrestrial ecoregions dataset[69]. We did this based on the calibration area or 'M element' of the BAM diagram that refers to areas that have been accessible to the taxon via dispersal over relevant periods of time[91,92]. We randomly sampled 10,000 background localities from the selected areas.

To reduce model complexity without compromising model performance, we built several models by varying the feature classes (FC) and regularization multipliers (RM) (see refs. 93–95) using R 3.6.0[96] and 'ENMeval' version 0.3.0 package[97]. FC determines the flexibility of the modeled response to the predictor variables, while the RM penalizes model complexity[93]. Occurrence records were randomly divided into 70% for model selection, and 30% of data was withheld for model validation. ENMeval carries out an internal partition of localities to test each combination of settings. Therefore, we selected the random $k$-fold method to divide localities into four bins. We build models with six FC combinations and varied RM values ranging from 0.5 to 4.0 in 0.5 increments. Optimal models were selected using Akaike's Information Criterion corrected for small sample sizes ($\triangle AICc = 0$). This method penalizes overly complex models and helps to choose those with an optimal number of parameters. However, it has been shown that the number of model parameters may not correctly estimate degrees of freedom[98], and that model selection should not be selected solely with one measure[99]. Thus, we used 30% of the withheld data to test the area under the curve (AUC) of the receiver operating characteristic, and the omission error under a 10 percentile training threshold.

We used the ten percentile or minimum training presence threshold to obtain binary maps of the presence and absence of suitable areas for species distribution. We asked experts of each taxonomic group who were also involved in the extinction risk assessment to select one of these two options and to indicate possible overestimated areas, which were then eliminated case by case using the information of Mexican ecoregions[100] and watersheds[101]. Eight models were binarized with the minimum training presence threshold; for the other models we used the 10 percentile threshold. See MaxEnt performance and significance of SDM at Supplementary Data 5. AUC values ranged from 0 to 1; 0.5 indicated a model performance not better than random, while values closer to 1 indicated a better model performance; here we used SDM showing AUC values higher than 0.7. For *Phaseolus* and *Zea*, we used SDM that were previously generated by Delgado-Salinas et al.[102], and Sánchez González et al.[103], respectively. SDM for 116 taxa were validated by experts of each taxonomic group. See references and download links at Supplementary Data 6.

For the conservation planning analysis of Mexico, we clipped the models to the Mexican territory, and trimmed the continuous SDM using the binary SDM to keep pixel values of areas with elevated probability of taxa presence. For taxa without SDM, we included the occurrence records of these taxa in the spatial analysis by using the information on observation location, i.e., coordinates (see Supplementary Data 3). This is done by enabling the function 'species of special interest' (SSI). See further details in the method section 'Final conservation analysis'.

## Proxies of genetic differentiation

To identify proxies of genetic differentiation in an explicit, efficient, and repeatable way, we included environmental and historical drivers of genetic diversity. For this, we first divided Mexico into 27 Holdridge life zones (Supplementary Fig. 2, Supplementary Data 8), which we then subdivided according to phylogeographic studies that have found genetic differentiation among populations of several taxa (see division of each life zone into proxies in Supplementary Fig. 4; Supplementary Fig. 3 provides a general geographical overview of Mexico and main geographic references mentioned in Supplementary Fig. 4). The literature review was done searching for the words "phylogeography" and one of the following: (i) name of the Mexican biogeographic zones, (ii) "Mexico" + an ecosystem name (e.g. "Mexico" "rainforest") or (iii) "Mexico" + lowlands/highlands. See list of references used in this study in Supplementary Data 9.

In addition, we manually reviewed the citations to the most cited papers of the previous search. Reviews and meta-analyses were also included, although we excluded studies performed in CWR to show that our approach can be used without prior information on this group. As more studies on such taxa become available, they can be used to fine-tune the proxies of genetic differentiation. We focused on terrestrial species including plants, animals, and fungi (Supplementary Data 10) except to subdivide a life zone covering the coasts of the California Peninsula, where we could not find studies on terrestrial taxa so we included studies on fish species (see Supplementary Fig. 4).

Since most of the life zones cover large territories, and complete phylogeographic congruence among different taxa is uncommon, we targeted to represent general trends that would likely occur across diverse species, instead of trying to represent fine idiosyncratic patterns of genetic differentiation. For instance, although distribution ranges of highland taxa shifted during the Pleistocene climate fluctuations, in general populations persisted (glacial-interglacial periods) within the main mountain ranges, while lowland populations were ephemeral (only glacial periods). So, gene flow among mountain ranges was more limited than within them. As a result, genetic differentiation among mountain ranges of different biogeographic provinces has been widely documented[32], so we used this general pattern to subdivide the life zones that occur in highlands. These types of patterns are particularly relevant for a country like Mexico, due to its complex topography, tropical latitude, and geographic features of different ages, which promote population differentiation among the Mexican main geographic features. To translate the phylogeographic information into a spatial context, we used biogeographic regions, basins, topographic or edaphic data to split the life zones into different subzones using the best fitting cartography to represent the phylogeographic patterns (Supplementary Fig. 4).

We obtained 102 proxies of genetic differentiation for Mexico (Supplementary Fig. 5). We validated our findings by using available genomic data of an empirical study of a wild relative of maize, the teosinte *Zea mays* subsp. *parviglumis*, which was not included in the literature review in order to test the usefulness of our approach regarding the lack of genetic data. The dataset includes ca. 1800 occurrence records and ca. 30,000 SNPs[48]. Sampling localities were not used for distribution modeling. Admixture groups per population were estimated for K1 to 60. According to the population analysis, *Z.*

*mays* subsp. *parviglumis* is structured in 13 genetic clusters along a longitudinal gradient (Fig. 3a–c). We used the K = 13 for plotting based on the Cross-Validation error. The proportion of each genetic cluster was estimated by sampling locality and plotted using pie charts over the map (Supplementary Fig. 6). Then, using the data layer of the SDM subdivided by proxies of genetic differentiation, we extracted which was the proxy most frequent in a 5 km buffer for each sampling locality. The Admixture plot was ordered by all genetic clusters and subdivided by the proxy of genetic differentiation most frequent for each locality. In addition, we calculated a principal component analysis (PCA) and projected into a score plot the first three components. Individual samples were colored by the proxies where they fell in the 5 km buffer (Fig. 3c). To compare how genetic variation was represented by the different scenarios we plotted the proportion of the area of each proxy as given by the potential SDM according to two different scenarios (only considering SDM; combining SDM*PGD) considering 20% of Mexico's terrestrial area (Fig. 3d). Analyses were run in R version 3.5.1[96] using the R packages pcadapt version 4.3.3[104], ggplot2 version 2_3.3.3[105], readr version 1.4.0[106], gridExtra version 2.3[107], ggnewscale version 0.4.5[108], scatterpie version 0.1.5[109], pophelper version 2.3.1[110], raster version 3.4-5[111], rgdal version 1.4-8[112], rgl version 0.107.10[113], and sp version 1.4-4[114,115].

## Habitat preference

We considered habitat preference to refine the presence of CWR in the planning process; thus minimizing commission errors and highlighting areas that more probably contain taxa[116]. For each taxon, experts assessed its habitat preference (1: high preference; 0.5: low preference; 0.1: no preference) according to the following categories: (i) well-conserved vegetation (i.e. primary vegetation), (ii) human-impacted vegetation (i.e. secondary vegetation), (iii) less intensive rainfed and moisture agriculture, (iv) intensive rainfed and moisture agriculture, (v) irrigated agriculture, (vi) induced and cultivated grasslands and forests, and vii) urban areas (Supplementary Data 11). To spatially delimit these classes, we used the land use cover and vegetation map for Mexico[117], and assessed seven main categories of land cover by grouping the map legend (Supplementary Fig. 9). To differentiate between less intensive and intensive cultivated areas, we followed Bellon et al.[56], who associated the presence of native maize varieties of Mexico to occur in municipalities with average yields of less than or equal to 3 t ha-1 using agricultural production data from 2010 from the Information System of Agrifood and Fisheries (SIAP), and selected the municipalities with the established average maize yield. We combined the municipality layer with the land cover map to differentiate areas of high and low agricultural intensity. To generate taxon-specific habitat layers, we associated the habitat preference classes established by experts to the land cover map aggregated into seven major land cover categories, using R 3.6.0[96] and the following packages: raster version 3.4-5[111] and rgdal version 1.4-8[112]. We obtained habitat maps for 116 taxa with SDM.

## Preliminary analysis

We generated five preliminary scenarios to explore different approaches to include conservation features for maximizing the representation of intraspecific diversity as given by taxa and proxies of genetic differentiation, i.e., representation of proxies within a taxa range (Supplementary Fig. 7): (i) "SDM" scenario, included 116 SDM, which we used as base scenario to examine the representation of taxa and proxies of genetic variability (n = 116); (ii) "SDM + LZ" scenario, included 116 SDM and 27 layers representing Holdridge life zones to consider environmental variation (n = 143); (iii) "SDM + PGD" scenario, included 116 SDM and 102 layers representing each proxy of genetic differentiation individually (n = 218); (iv) "SDM*PGD" scenario, included 5004 input layers representing the intersection of SDM and PGD (n = 5004; combining 116 SDM with 102 proxies resulted in 11,832

layers, but as some of the intersections produced empty outputs given the extension of SDM that do not cover all Mexico, for further analysis we used 5004 input layers with value data. To subdivide the layers, we used ArcGIS version 10.2.2[118]; to filter the layers, we used R 3.5.1[96].); (v) "SDM and PGD as ADMU" scenario, included 116 SDM as the main conservation features, while integrating one single layer of proxies of genetic differentiation to consider each of them as planning units by using the 'Administrative units' function. Analysis was done in Zonation[50,119].

We compared the results by assessing 20% of Mexico's terrestrial area (Fig. 5b) to perform statistical analysis in R 3.5.1[96] using the following packages: purrr version 0.3.4[120], 'dplyr' version 1.0.2[121], 'ggplot2' version 2_3.3.3[105], 'raster' version 3.4-5[111], 'scales' version 1.2.0[122], 'sp' version 1.4-4[114,115], 'tidyr' version 1.0.2[123], and 'vegan' version 2.6-2[124]. The area threshold was established based on Aichi target 11 and on comparisons of performance curves to efficiently represent taxa ranges delimited by SDM and proxies of genetic differentiation (Fig. 6). As using SDM combined with proxies of genetic differentiation showed the highest representation of genetic diversity ("SDM*PGD" scenario), we used this approach for the final analyses.

## Final conservation analysis

We identified areas of high conservation value for CWR in Mexico by using the software Zonation version 4.0[50,119], a systematic conservation planning tool that allows optimizing representation of species, taxa, or other conservation features, e.g., proxies of genetic differentiation, in a given study area. The program hierarchically ranks areas by removing cells of low conservation value, as given, for example, by a reduced number of taxa or occurrence of low weighted features, while considering multiple criteria such as the weighting of taxa and habitat preference of taxa. We applied the core-area zonation removal rule (CAZ) to maximize the representation of all conservation features in a minimal possible area[51]. Zonation generates two main outputs: (a) a hierarchical landscape priority rank map, that allows decision makers establishing different area thresholds to highlight areas of conservation interest; and (b) a representation curve showing species or conservation features range distribution in a given area. The curve also allows identifying how much area is needed to cover a certain taxon range or the distribution of a feature of conservation interest.

For the conservation scenarios, we integrated the following inputs in the Zonation software: (1) 5,004 layers, i.e., SDM intersected with proxies of genetic differentiation (as described by "SDM*PGD" scenario, Fig. 4), (2) occurrence records of 98 taxa; only for those taxa without SDM, see Supplementary Data 3), (3) taxa specific habitat layers (according to Supplementary Data 11 and Supplementary Fig. 9), and (4) IUCN threat category (Supplementary Data 3) as an additional parameter to weight taxa differently to consider their vulnerability to extinction, see details below. See Zonation configuration at Supplementary Note 6.

Data from different sources can be mixed in the same analysis, which is useful to not lose or omit information of any taxa of interest in the assessment. Here, we included information of a total of 214 taxa (see Supplementary Data 3). Distribution data of 116 taxa were represented by 5004 layers that resulted from combining 116 SDM and 102 PGD. This approach showed the highest proportion of area of taxa ranges (on average 41%) and highest representation of PGD within the area of each taxon (on average 76%; Fig. 4; see description in the main text). For some taxa, e.g. *Cucurbita pepo, Physalis cinerascens*, and *Zea mays* information on its distribution was assessed at subspecies level rather than at species level, explaining the difference in numbers of CWR taxa.

In addition, we included occurrence data of 98 taxa without SDM to prevent missing important areas of taxa known distribution that are important to conserve (see Supplementary Data 3). We enabled the function 'species of special interest' (SSI) of Zonation, and included a

SSI feature list file, listing the taxon names, as well as taxon-specific coordinate file for each of the 98 taxa that have been reviewed by the experts of each group. The spatial reference system was World Mercator projection. Occurrence data and SDM are treated similarly in the Zonation analysis, i.e., cells where taxa occur will be retained in the solution as long as possible to maximize its representation in the solution.

We assigned weights to the 116 taxa with SDM by using IUCN threat categories (according to Supplementary Data 3), giving highest values to taxa with highest risk of extinction that urgently need management actions to further avoid genetic erosion. By including conservation feature weights, Zonation estimates the conservation value of a cell not only based on the presences of a taxa and their distribution range, but also on the weight. A high weight indicates a high conservation value of cells where these taxa are distributed. As there is no rule for weight setting, we assigned values between 1 and 0 regardless of taxa distribution ranges, which is automatically considered in the Zonation algorithm to guarantee the representation of locations where limited-range distributed taxa occur within the most valuable conservation area. Thus, weights were assigned as follows: Critically endangered, CR: 1; Endangered, EN: 1; Vulnerable, VU: 0.8; Near threatened, NT: 0.5; Data deficient, DD: 0.3; Least concern, LC: 0.2 Not evaluated, NE: 0.1. SSI taxa were all weighted similarly with 1 in order to represent the 98 SSI taxa and their occurrences in the top fraction of the most valuable conservation area, as these areas could be considered as 'irreplaceable' in terms of conservation. The conservation of these taxa that are only known in a few locations is crucial to maintain their populations. Information on weights for taxa with and without SDM is included in the file that lists the 5004 conservation features and the SSI file, respectively.

To include the information on habitat, we included 116 habitat maps which guide the selection of cells to areas where its presence is more probable (see the Methods section: "Habitat preference"). This option can only be used for taxa represented by a raster layer, and is not available for SSI taxa included via occurrence records. By enabling the "landscape condition" option of Zonation, each habitat map is linked to a specific conservation feature layer. Areas with unfavorable habitats will quickly be masked out during the selection of cells in order to obtain a solution that favors conservation areas within areas of preferred habitat.

We generated three final scenarios to identify conservation areas for (a) all taxa, (b) taxa exclusively distributing in natural vegetation, and (c) taxa associated with a wider range of habitats such as natural vegetation, agricultural and urban areas. The Zonation configuration remained similar among the three scenarios. When taxa were not included in a given scenario, we assigned a value weight of 0. This excluded the feature to be considered for the hierarchical prioritization of the landscape, but still allowed to evaluate the taxa during post-processing.

To evaluate the spatial results (Supplementary Fig. 11), we analyzed performance curves to represent proxies of genetic differentiation within each taxon range (Supplementary Fig. 12). Also, we considered the most valuable 20% area of Mexico to calculate the coincidence of the three scenarios (Supplementary Fig. 13), and the overlap with federal protected areas[125] and indigenous regions[126,127] (Supplementary Fig. 14), and land cover data used in the analyses (Supplementary Figs. 9, 15).

We discussed the proposed methodological framework, input layer and criteria during a fourth workshop in Mexico. It is worth mentioning that we ran several analyses including additional layers, such as areas where indigenous communities live that promote the presence of CWR in the landscape[6]. However, as the output indicated no evident difference by including this information, final analyses did not consider these data. We neither included protected areas nor tried to expand on the current 12% protected area system, because most

management plans do not specifically address CWR management (but see the management program of the Protected Area of 'Sierra de Manantlán'[128]), and thus generally do not adequately plan for wild and native genetic resources[129]. We also discussed different approaches to consider connectivity for taxa, habitats and proxies of genetic differentiation in the Zonation processing. Still, we finally decided to run the analysis without particularly accounting for connectivity as we had no taxa-specific information on dispersal abilities or possible effects of fragmentation, and we did not want to lose efficiency of the solution to represent taxa by or include lower-quality habitats by forcing the solution to an aggregation of pixels.

### Reporting summary

Further information on research design is available in the Nature Research Reporting Summary linked to this article.

## Data availability

The Zonation input and output files, including potential species distribution models (SDM), occurrence records, table with IUCN category per taxa, habitat rasters, genetic data and metadata used in this study are available in Dryad under https://doi.org/10.5061/dryad.7m0cfxpxm[130]. SDM can also be downloaded at Conabio's GIS portal (http://www.conabio.gob.mx/informacion/gis/; and Supplementary Data 6 for direct download links). Source data to make Figs. 3, 4, 6 are provided with this paper. Source data are provided with this paper.

## Code availability

Custom R scripts and Zonation files used for the analyses and figures of this study are available at the Github repository https://github.com/CONABIO/analisisUniCons_proxiGen[131]. Zonation settings are also available in Supplementary Note 6.

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

## Acknowledgements

We are grateful to the Darwin Initiative of the United Kingdom for providing funding, and to the International Union for Conservation of Nature, IUCN, for implementing the project "Safeguarding Mesoamerican Crop Wild Relatives" (project number: 23-007) granted to IUCN (B.G.) and CONABIO (P.K., F.A.G. and T.U.-H.). Analyses in this paper were carried out on CONABIO's computing cluster, supported by their system administrator and the Subcoordinación de soporte informático. We thank participants who collaborated during the Darwin Initiative project and our colleagues from CONABIO who provided technical support: Jorge Acosta Gallegos, Alejandra Barrios Pérez†, Juan Barrios Vargas, Ernesto Campos Murillo, Jamie A. Carr, Gabriela Castellanos-Morales, José G. Cerén López, Aremi R. Contreras Toledo, Nancy Corona Pedroza, María Eugenia Correa-Cano, Moisés Cortés Cruz†, Lino de la Cruz Larios, J. Fernando de la Torre, Daniel G. Debouck, Cuauhtémoc Enríquez García, Gloria Espinosa Sánchez, Patricia Galindo Hernández, Oscar Godínez Gómez, Emma Gómez Ruiz, Enrique González-Pérez, Abraham Guerrero, Mariana Hernández-Apolinar, Braulio E. Herrera-Cabrera, Lev Jardón Barbolla, Megan Jefferson, Mahinda Martínez, Jenny Menjívar, María de los Ángeles Mérida Guzmán, Aura J. Morales Herrera, Albaro Orellana, Daniel Ortiz Santamaría, Mario Parada Jaco, Caroline M. Pollock, Diana Ramírez Mejía, Guillermo Sánchez-de la Vega, Mariella Superina, Marcel F. Tognelli, Heike Vibrans, and Pilar Zamora Tavares. Special thanks to Richard K.B. Jenkins, Nigel Maxted and Shelagh Kell for comments to our approach and their contribution to the Darwin Initiative project. We are thankful to Andrés Lira Noriega, Daniel Piñero and Jorge Soberón for feedback on the manuscript.

## Author contributions

A.M.-Y., W.T.-N., T.U.-H., B.G., F.A.G. and P.K. conceptualized and designed the methodology. W.T.-N., A.M.-Y., A.P.C.-R. and D.R.-R. performed analyses. W.T.-N., B.G., A.P.C.-R., O.O.-G., M.A.O.-R., E.U.-H. and J.A.-G. curated data. P.K., T.U.-H., F.A.G., M.A.O.-R., W.T.-N., A.M.-Y., B.G., E.U.-H., O.O.-G., A.P.C.-R., J.A.-G. and C.B. coordinated workshops and organized experts' contributions. F.A.C., D.R.-R., J.J.S.G., A.A.-M., V.A., G.A.-I., C.H.A.-A., C.A.P., A.D.-S., P.G., M.G.-L., J.H.-R., F.G.L.-H., R.L.S., A.R., D.R.D., J.A.R.-C., J.J.S.P., O.V.-P., M.V., A.W. and M.Q.-C. provided data, participated in workshops and validated models. B.G., J.S., P.K., F.A.G. and T.U.-H. acquired funding and envisioned the project. W.T.-N., A.M.-Y. and T.U.-H. lead writing with contributions of P.K., F.A.G., B.G., A.P.C.-R., E.U.-H., M.A.O.-R. and O.O.-G. All authors reviewed, commented and approved the manuscript.

## Competing interests

The authors declare no competing interests.

## Additional information

Article

Wolke Tobón-Niedfeldt ®[1], Alicia Mastretta-Yanes ®[1,2] ✉, Tania Urquiza-Haas[1], Bárbara Goettsch[3,4],
Angela P. Cuervo-Robayo ®[1], Esmeralda Urquiza-Haas[1], M. Andrea Orjuela-R[1], Francisca Acevedo Gasman[1],
Oswaldo Oliveros-Galindo[1], Caroline Burgeff[1], Diana M. Rivera-Rodríguez ®[5], José de Jesús Sánchez González[6],
Jesús Alarcón-Guerrero[1], Araceli Aguilar-Meléndez[7], Flavio Aragón Cuevas[8], Valeria Alavez[9,10],
Gabriel Alejandre-Iturbide[11], Carlos-H. Avendaño-Arrazate[12], César Azurdia Pérez[13], Alfonso Delgado-Salinas[14],
Pablo Galán ®[15], Manuel González-Ledesma[16], Jesús Hernández-Ruíz ®[17], Francisco G. Lorea-Hernández[18],
Rafael Lira Saade[19], Aarón Rodríguez[6], Dagoberto Rodríguez Delcid[15], José Ariel Ruiz-Corral[6], Juan José Santos Pérez[20],
Ofelia Vargas-Ponce[6], Melania Vega ®[9,10], Ana Wegier[9], Martín Quintana-Camargo[21], José Sarukhán[1,22] & Patricia Koleff[1]

[1]Comisión Nacional para el Conocimiento y Uso de la Biodiversidad (CONABIO), Mexico City, Mexico. [2]Consejo Nacional de Ciencia y Tecnología (CONACYT),
Mexico City, Mexico. [3]Cactus and Succulent Plants Specialist Group, Species Survival Commission, International Union for Conservation of Nature (IUCN),
Cambridge, UK. [4]The Biodiversity Consultancy Ltd, Cambridge, UK. [5]Departamento de Ciencias Básicas, Instituto Tecnológico de Tlajomulco, Tecnológico
Nacional de, México, Jalisco, Mexico. [6]Centro Universitario de Ciencias Biológicas y Agropecuarias (CUCBA), Universidad de Guadalajara, Zapopan, Mexico.
[7]Centro de Investigaciones Tropicales, Universidad Veracruzana, Veracruz, Mexico. [8]Instituto Nacional de Investigaciones Forestales, Agrícolas y Pecuarias
(INIFAP), Campo Experimental Valles Centrales, Oaxaca, Mexico. [9]Laboratorio de Genética de la Conservación, Jardín Botánico, Instituto de Biología,
Universidad Nacional Autónoma de México (UNAM), Mexico City, Mexico. [10]Posgrado en Ciencias Biológicas, UNAM, Mexico City, Mexico. [11]Centro Inter-
disciplinario de Investigación para el Desarrollo Integral Regional, Unidad Durango, Instituto Politécnico Nacional, Durango, Mexico. [12]INIFAP, Campo
Experimental Rosario Izapa, Chiapas, Mexico. [13]Consejo Nacional de Áreas Protegidas, Guatemala, Guatemala. [14]Instituto de Biología, UNAM, Mexico
City, Mexico. [15]Asociación Jardín Botánico La Laguna, Herbario LAGU, San Salvador, El Salvador. [16]Herbario HGOM, Centro de Investigaciones Biológicas,
Instituto de Ciencias Básicas e Ingeniería, Universidad Autónoma del Estado de Hidalgo, Hidalgo, Mexico. [17]Universidad de Guanajuato, Guanajuato, Mexico.
[18]Instituto de Ecología, A. C., Xalapa, Veracruz, Mexico. [19]Laboratorio de Recursos Naturales, UBIPRO, Facultad de Estudios Superiores Iztacala, UNAM, Mexico
City, Mexico. [20]Instituto de Ciencia y Tecnología Agrícola, Guatemala, Guatemala. [21]Centro Nacional de Recursos Genéticos, INIFAP, Tepatitlán de
Morelos, Mexico. [22]Instituto de Ecología, UNAM, Mexico City, Mexico. JAG's Current position: independent. ✉e-mail: amastretta@conabio.gob.mx

