## [Peer Review File · Nature Communications]

Reviewers' Comments:

Reviewer #1:

Remarks to the Author:

The authors have combined a huge amount of data to provide conservation management recommendations for crop wild relatives in Mexico. This is clearly a massive effort, and the authors have put together some clear conservation recommendations, including specific regions that could be prioritized. I congratulate the authors on such a comprehensive analysis, integrating so many types of data, and their clear results.

I appreciate the challenge of fitting such a large study into a relatively short format, but I felt as though there were many key details missing, including in the supplementary methods and results. See the specific comments below for the places where I wanted a bit more detail. I also recommend removing acronyms to improve understanding, especially acronyms that are not used much throughout the paper (e.g., PA).

Especially given the necessity for brevity in the Nature Comms format, I felt that the introduction could have been better focused. Specifically, despite telling me that crop wild relatives are important, I felt it lacked the explanation to convince a reader that crop wild relatives are of particular interest from a conservation perspective. I recommend revisiting this section to highlight the importance of crop wild relatives and why we need to also consider genetic diversity. To that end, I thought that the discussion could have also discussed how/whether the results could be used to inform conservation genetic methods (e.g, translocations).

I thought it was excellent that the maize relatives were used to validate the models, however I'm not sure that the results show that using the proxies of genetic diversity is effective. Based on Fig. 3a, the groupings based on the proxies of genetic diversity do not seem to line up with the actual genetic clusters -- though they do seem to match up somewhat in Fig. 3b, so perhaps Fig 3a is misleading. Either way, these results need to be explained in more detail and presented in a more interpretable way. Also, the admixture plots show quite a lot of admixture, with patterns that could almost be isolation by distance, which could make the admixture results somewhat suspect. I recommend the authors complement the admixture analysis with another demographic modeling approach, for example a PCA-based (e.g., PCAdapt, <https://bcm-uga.github.io/pcadapt/articles/pcadapt.html>) or maximum-likelihood based approach (e.g., Treemix, <https://bitbucket.org/nygcresearch/treemix/wiki/Home>).

Specific comments:

Line 86: Is biodiversity actually persisting in mosaic landscapes? Consider rephrasing to emphasize instead that biodiversity is challenged by mosaic landscapes.

Lines 87-88: second sentence not necessarily a good flow -- consider removing (it's essentially reiterated in line 92).

Lines 92-93: not sure why crop wild relatives are particularly urgent from this sentence, consider stating more explicitly why these are important for food security.

Lines 105-110: Is the main risk gene flow from crops -> wild relatives, or is there a risk (or perceived risk by growers) of wild relatives -> crops?

Lines 111-112: Need to explain the planning based on the Magos Brehm methodology -- this is not a commonly known method.

Lines 120-124: The authors could more clearly state which aspects of the framework are novel, as they also are relying on the so-called Magos Brehm framework.

Lines 131-136: I recommend referring to the supplementary methods in here somewhere to direct the reader to a more detailed description. Also, consider re-phrasing to use less jargon-filled

terminology, for example replacing 'CWR checklist' with 'creating a list of focal crop wild relative species'.

Lines 132-133: The focus of the manuscript and analysis seems to be on in situ conservation, rather than ex situ conservation -- but perhaps I have misunderstood.

Lines 186-190: I'm not sure I completely understand what 76% of proxies for genetic diversity mean -- the figure makes it seem as though this might be a proportion of a geographic distribution, but that is not how I would have interpreted a proxy of genetic diversity. I would have expected a proxy of genetic diversity to refer to some level of within-taxon variation or divergence. I recommend that the authors add more detail to the description of these metrics and results.

Line 200 and line 216: I was unfamiliar with the Aichi Target 13 of CBD -- consider using the entire name (Convention on Biological Diversity) and providing a link/reference.

Lines 199-207: This context seems more appropriate for the discussion than the results.

Line 217: I wanted a bit more detail on what the authors meant by 'considering 20% of Mexico' -- did this 20% include already-protected areas? How were the 20% of terrestrial regions chosen? What was maximized?

Lines 248-262: To me, this seemed like it belonged more in the discussion than in the results.

Lines 278-303: This section also seemed to belong in the discussion rather than the results.

Fig. 1: the text in the figure is a bit too small. Also, I got confused by the spatial representation of phylogeographic patterns split into biogeographic provinces, edaphology, and watersheds -- from reading the methods, I thought the phylogeographic patterns came from a literature review of genetics-based phylogeography. The methods for this step need to be more clearly conveyed.

Fig. 5a: Unclear what the continuous map in (a) is showing -- what is 'accumulated area'? Please provide more detail in the caption, and consider a more descriptive label in the figure.

Fig. 5b: This is said to be considering 20% of Mexico's terrestrial area -- but considering it for what? The caption should be more explicit about what is being presented. It's somewhat unclear whether the brown areas are overlaid over some of the green and brown areas (or whether they are simply neighboring each other), which would be important to know (i.e., whether the conservation areas are already in protected areas). The authors also might consider including highly urban areas on the map as well.

Fig. 6: What does it mean to have the proportion of PDG area? Consider adding more context in the caption to help with interpretation of these curves.

Supplementary Methods

In the methods overview, please refer to the sub-sections that follow to indicate to the reader that more detail is available below.

Note: the plural of genus is genera.

CWR checklist section:

The first set of criteria are unclear on what level they were applied to -- presumably not species level, since they are followed by criteria 4 and 5 for species-level. Please clarify this.

What does it mean for a species to be included in primary, secondary, or tertiary gene pools? This terminology is not clear, please elaborate.

These methods describe each country compiling a list, but the previous section stated that the analysis is focused on Mexico -- which is it?

The 224 taxa in the final inventory are all related to the crop species, but do not include the crop species themselves -- is that correct? The wording is a bit unclear.

Species distribution models:

How did the AUC test used to inform model selection? Did it always agree with AIC, or did the AUC test impact the model that was chosen?

"Experts selected the best fitting SDM" -- which experts? I would expect the experts to be part of the research team writing this paper, was that not the case? How did they validate the SDMs?

Did using 116 SDMs mean that the number of species used was reduced from 224 to the 116 with SDMs?

Proxies of genetic diversity:

I have a few questions about the literature review that was used to characterize spatially explicit phylogeographic patterns:

- Did this review focus only on species with similar dispersal mechanisms to the crop wild relatives studied (e.g., I would expect lake fish to be structured very differently from wind-dispersed plants)?

- How much variation was there among the species? Did they all show similar phylogeographic patterns, or was it very species-specific? This seems critically important to assess how well the genetic diversity proxies might work.

The text states that of the 11832 GIS layers created, only 5004 were used -- how were these selected?

Spatial conservation planning analysis:

In the paragraph describing the final conservation analyses, refer to the numbered approach from the previous paragraph that was chosen.

Supplementary results:

The supplementary figures could do with more descriptive figure captions.

Taxa distribution patterns:

The text states that the areas of highest richness are those with the highest sampling effort -- have the authors considered rarefaction analysis to evaluate whether sampling effort has been sufficient to estimate species richness?

Reviewer #2:

Remarks to the Author:

This paper develops a novel way to incorporate within-species' genetic diversity/ evolutionary resilience into systematic conservation planning (i.e. data-driven ranking of what areas should be protected for maximizing biodiversity conservation). To my knowledge, the approach- with proxies of genetic data (PGD)- has not been used before. Although genetic data has been considered recently in a few studies, research up to now has only included at most approximately 10 species, and generally required genetic data (e.g. Hanson et al 2021 Cons Biol). Other research has assumed that percent of species area equates to a good proxy of genetic diversity (Khoury et al 2020 PNAS). This paper is a major advance on such earlier work because it explicitly includes phylogeographic divisions (from past genetic studies), but not actually requiring genetic data. Thus the approach can be applied to any number of species at national or even global scales. The authors compare the performance of their approach to alternatives such as only using species distribution models, and calculate both the percent of species' ranges and percent of PGD conserved. They also experiment with including other information as weights (IUCN status for

example) and other data including species' habitat preference and land use. I think it is a major contribution to the literature, and is highly timely for conservation policy (for example for the Convention on Biological Diversity and Sustainable Development Goals).

I have numerous requests for clarification about methods. I have other requests for clarification because the authors have evaluated many different scenarios and it is a bit hard to follow. However, I do not see any major flaws in the work.

Larger requests

Methods- line 164- How many species were included in this literature review? Were they all non CWR? Do they have similar characteristics, geographic range, etc. as crop wild relatives? In other words are they well representing CWR genetic 'breaks'?

Methods/ Results- line 164- It is not clear to me if phylogeographic breakpoints were actually used. It looks like in Fig 1 that divisions such as watersheds and soil were used. But was there any method used to actually include breakpoints indicated by genetic data (attempts have been made in N America for this, e.g. Soltis et al 2006, Shafer et al 2010)? If not, the text should be more clear that phylogeographic genetic datasets were not used explicitly but rather generally support breakpoints at watersheds etc.

Methods/ Results- line 186 onward- Even after reading the Supplemental its hard to picture the difference between PGD+SDM and PGD*SDM. Please explain more. Also, why not include a PGD only scenario (just PGD).

Methods/ Results- line 213 onward- I'm not sure how taxon specific habitat preferences are incorporated. Are some habitats 'masked out' or removed from the SDM? Or downweighted pixels? It is also not clear how occurrence records are incorporated into the Zonation procedure? It would also be great to read a few sentences on how many occurrence records are sufficient for a species. Results- It looks like the preliminary analysis resulted in a max of 76% of PGD per taxa (line 186) while the final analysis (including IUCN ranking etc) resulted in only 50% of PGD per taxa (line 237). Why the major drop?

Higher level- because there are so many scenarios analyzed, it can get confusing to follow the methods- a flow chart listing them all would be useful- the main Fig 1 only looks at the preliminary scenarios. A flowchart showing what goes into each scenario would be great- and matching that to Figures. Could be Supplemental.

A higher level discussion point/ critique. I think that selection of areas using Zonation will not consider connectivity, e.g. cells selected are at 1 square km (or whichever resolution) which could lead to selection of very tiny parcels of land which are isolated. Could you discuss this problem briefly? Such a prioritization would also not consider if the habitat (for example one square km) is large enough to support a viable population (such as effective size 500), which is also an indicator of genetic diversity. This might mean that genetic diversity is not sufficiently conserved!

A suggested analysis, if its possible- how many taxa have their PGD conserved at very low levels, say less than 60%? It looks from Fig 4 that very few... but which are they, and why? In other words why are some taxa PGD conserved very low- are they all in the same genus, spatial location, have similar biology? Is there anything in common to them? Also how many have >90% PGD conserved? The current draft of the CBD post 2020 framework (released today) suggests 90%.

Smaller requests

Line 112 briefly define what you mean by "Magos Brehm et al methodology"

Methods/ Results- SDM- what probability of occurrence was used as a threshold to (from Supp Matt) "trimmed continuous SDM using binary SDM"?

Methods/ Results- line 175- is there any way to better measure "no complete coincidence" between genetic and PGD data for maize? It looks like it was only compared "by eye"? Also is this a crop wild relative or an actual crop? Could other datasets be used for validation (e.g. those with chloroplast or microsat data)?

Figure 6 has PDG instead of PGD on y axis

Methods/ Results- line 272- Supp fig 8... these panels do not look very different to me- I think they look very similar. Can you point out some differences?

Discussion line 316- "this can inform public policy regarding living modified organisms such as crops and agriculture subsidies in order to mitigate threat processes to CWR." To me this seems

like a bit of a stretch, or maybe is just not clear to me. Can you explain?

Reviewer #3:

Remarks to the Author:

This article provides a thorough analysis to identify priority areas for in situ conservation of crop wild relatives in Mexico. It also provides a methodological approach to conduct such prioritization that incorporates evolutionary and threat assessment information. The article is the result of a large international multidisciplinary collaborative project and contains very relevant data and analyses related to crop wild relative conservation in Mexico. The results, i.e., the identification of priority areas for in situ conservation of CWR in Mexico, are noteworthy, especially given the fact that Mexico contain some of the most relevant hotspots of CWR diversity in the world. These results certainly add great information to the current knowledge of CWR diversity in Mexico. Having indicated this, the manuscript as it stands, is, in my opinion, at an early immature stage and has deficiencies concerning its purpose, focus and delimitation. Both the title and the last paragraph of the introduction suggest that the purpose of the manuscript is to present a new methodological approach for crop wild relative conservation that includes evolutionary and threat assessment information in the process. However, there is nothing of great relevance in this methodological approach. As indicated at the beginning of the Results section it basically follows the main steps of Magos Brehm et al (2017, 2019) proposals. The proposal does present a significant addition which involves the use of phylogeographic studies information as a proxy of the demographic history of the target CWR species. This way, in the classification of the territory that is going to be used as a proxy of within-species genetic diversity, in addition to the use of Holdridge life zones to account for differential environmental conditions, the authors use a synthesis of phylogeographic studies to account for the genetic diversity among populations that may have been generated through the past history of the species. Surprisingly, if this is the novelty of the study, and the purpose of the article is to present it as such, the manuscript provides no further information on how this is done other than "Then each life zone was divided according to phylogeographic patterns of non CWR species, using biogeographic regions or topographic and edaphic data to define the cutline". The authors provide a list of 42 references of phylogeographic studies that were used to conduct this process. Concerning this part of the methodology, there is not enough detail provided in the methods for the work to be reproduced. I think it is a good idea to try to incorporate some criteria into the classification of the territory that is going to be used as a proxy of genetic diversity that takes into account the demographic history of the target CWR. However, a) I do not think it is possible to correctly infer genetic diversity associated to the demographic history of the target CWR by extrapolating results from other life forms that include birds and reptiles, b) in any case, the authors give no information or discussion to support their claims, c) they neither provide a detailed description of how this partition of Holdridge life zones was done using the information resulting from the phylogeographic studies, d) they neither discuss how this approach can or cannot be applied to similar territories in other parts of the world.

Apart from this, the contents of the Results section are fuzzy. If the purpose of the article is to provide a new methodological framework, the Results section provides little clear description of what the new framework is other than what is depicted in Figure 1. Even the description provided in its legend is minimal. If there are other novelties in the framework (e.g., the comparative analyses of target CWR taxa depending on the nature of their habitat, the use of the Zonation software) than the use of evolutionary information, they should be clearly identified, described and presented in a first presentation of the methodology.

On the contrary, the Results section is mainly dedicated to present a detailed account of the results of implementing this methodological framework in a selection of target CWR for Mexico. In this sense, I feel that the value of this work precisely resides in these results because as I previously mentioned, I think that the methodological innovation concerning the use of evolutionary information is not sound and it is not properly explained. However, when it comes to this, there is an additional problem: the results of implementing steps I, II and III of the methodology have already been submitted for publication (Goettsch et al., in prep.), but they are not published. The authors provide a copy of the manuscript that has been sent to the journal *Plants People Planet*. There is no duplication of its contents, but the fact that the article has not been published makes it an unreliable reference (most journals only accept references concerning

published information). Furthermore, in this line, citation number 47 (Urquiza et al) is also a reference to a document that has not yet been published.

The Species Distribution Models have been performed rigorously and are described in great detail. One key result in the implementation of the methodology for Mexico is the generation of the proxies of genetic diversity (PGD). The authors claim partial support of their PGD by contrasting them with empirical data from the teosinte *Zea mays* spp. *parviglumis* obtained from a doctoral thesis (Figure 3). There is certainly an association between the genetic clusters obtained through molecular analysis and the corresponding PGDs. It is a pity that the results of the doctoral thesis are not yet published in a scientific journal (or at least there is no indication of this in the manuscript) to provide the possibility of further verification or analyses.

Some other parts of the implementation of the methodology include some additional analysis that may be of interest if its relevance is properly explained and discussed, but otherwise they contribute to the loss of focus. This is the case of the three independent analyses performed to different sets of CWR depending of their habitat preference. In the way they are currently presented they rest clarity and focus to the presentation of the results. Are these tests relevant? Do they really contribute to the purpose of this article? What is the use of the results presented in Supplementary Table 7 and Figs 6, 7, 10 and 11 in the context of the purpose of this article? What is their practical application? This is also the case of the tests of different alternatives to incorporate PGD into the spatial analysis. It would be clearer and more focused if Figure 4 would only present the results corresponding to the alternative finally used for this study (leaving the rest, or the comparison, in a supplementary Figure).

Another formal deficiency that needs to be improved concerns the assignation of contents to Results, Discussion and Methods. Depending on whether the article is focused as a methodological approach or as a conservation assessment of target CWR in Mexico, the Results section should be streamlined moving some of its contents to the Methods and Discussion sections. It is noteworthy that the Results section contains considerable interpretation of the results and association to other references that would probably fit better in the Discussion section. In this sense, the current Discussion section made out of just two paragraphs connects more with the idea of some concluding remarks than a proper discussion of the results.

In conclusion, there is a need to clarify the purpose of this article. I think the results of the assessment of CWR conservation in Mexico are much more relevant than the novelty of the methodology and I would suggest that the article is reoriented in this way. In any case, the contents of the manuscript must be properly developed in the corresponding Methods, Results and Discussion sections, and only the information that is necessary to support the main purpose of the article should be included, eliminating other information that is less relevant. Citations to unpublished references should be eliminated if possible and the relationship with information about other parts of the project should probably be dealt in a different way.

Incorporating evolutionary and threat processes into crop wild relatives conservation
(tracking number: NCOMMS-21-17838A).

Reply letter to Reviewers comments

Dear editor,

We greatly appreciate the advice given by the reviewers and you to our previous submission. We have carried out the corrections they suggested. In particular, we have carried out the additional analysis suggested by Reviewer 1 (PCAdapt) and we have made available for review the dataset requested by Reviewer 3 (which will be also available in a public repository upon acceptance, as now stated in our Data availability section). We also have extended the details of our Methods both in the main text and in the Supplementary Information, addressing all the reviewer's questions. We also included a new Supplementary Information, where the methods and discussion of the proxies of genetic differentiation are fully detailed, which was another of the major concerns of the three reviewers.

Below we provide point-by-point responses in blue text. Changes made are highlighted in green in the main text and the supplementary materials files.

Dear Dr Mastretta-Yanes,

Thank you again for submitting your manuscript "Incorporating evolutionary and threat processes into crop wild relatives conservation" to Nature Communications. We have now received reports from 3 reviewers and, after careful consideration, we have decided to invite a major revision of the manuscript.

As you will see from the reports copied below, the reviewers raise important concerns. We find that these concerns limit the strength of the study, and therefore we ask you to address them with additional work. Without substantial revisions, we will be unlikely to send the paper back to review. In particular, we expect the additional analysis suggested by Reviewer 1 and the additional data suggested by Reviewer 3. Please note that you have wide margins to expand the main text (i.e. Introduction, Results, and Discussion, preferably max 5,000 words), and the Methods may be as long as required to address the reviewers' questions since the Methods section is not included in the word count.

If you feel that you are able to comprehensively address the reviewers' concerns, please provide a verbatim point-by-point response to these comments along with your revision. Please show all changes in the manuscript text file with track changes or colour highlighting. If you are unable to address specific reviewer requests or find any points invalid, please explain why in the point-by-point response.

Reviewer #1 (Remarks to the Author):

The authors have combined a huge amount of data to provide conservation management recommendations for crop wild relatives in Mexico. This is clearly a massive effort, and the authors have put together some clear conservation recommendations, including specific regions that could be prioritized. I congratulate the authors on such a comprehensive analysis, integrating so many types of data, and their clear results.

I appreciate the challenge of fitting such a large study into a relatively short format, but I felt as though there were many key details missing, including in the supplementary methods and results. See the specific comments below for the places where I wanted a bit more detail. I also recommend removing acronyms to improve understanding, especially acronyms that are not used much throughout the paper (e.g., PA).

Reply 1: Thank you for the observation. We made major edits throughout the document (see replies 32, 33, 38, 39, 55-57 for detailed changes and line numbers) and SI. We also included a longer version of the methods in order to highlight the novelty of our approach and complement missing key information, including a new Supplementary Information (SI 3: Notes) where we detail the PGD methodology.

Also, we have removed unnecessary acronyms (PA, CBD, AVDC, SCP). Please notice that PGD now stands for “proxies of genetic differentiation”. See reply 13 for details.

Especially given the necessity for brevity in the Nature Comms format, I felt that the introduction could have been better focused. Specifically, despite telling me that crop wild relatives are important, I felt it lacked the explanation to convince a reader that crop wild relatives are of particular interest from a conservation perspective. I recommend revisiting this section to highlight the importance of crop wild relatives and why we need to also consider genetic diversity. To that end, I thought that the discussion could have also discussed how/whether the results could be used to inform conservation genetic methods (e.g, translocations).

Reply 2: We have made major edits to the introduction (lines 101-113) to highlight the importance of CWR as a reservoir of genetic diversity for food and nutrition security under global change, and have stressed the need to conduct systematic conservation planning analysis that explicitly use criteria to consider genetic diversity for guiding conservation action, in particular for Mesoamerican wild relatives of some of the world’s most important crops

Regarding the discussion, as stated in the MS (lines 227-229, 375-380, 420-425), the PGD are important to identify areas for *in situ* and *ex situ* conservation when planning for biodiversity at the genetic level, which can furthermore support strategic actions, plans and programs to protect CWR. While PGD may inform on translocation of species, due to its complexity, we consider that discussion is out of the scope of this paper.

I thought it was excellent that the maize relatives were used to validate the models, however I’m not sure that the results show that using the proxies of genetic diversity is effective. Based on Fig. 3a, the groupings based on the proxies of genetic diversity do not seem to line up with the actual genetic clusters -- though they do seem to match up somewhat in Fig. 3b,

so perhaps Fig 3a is misleading. Either way, these results need to be explained in more detail and presented in a more interpretable way.

Reply 3: We have modified Fig. 3 to better show how the empirical differentiation relates to our PGD. For this, below the admixture plot we now show color bars matching the PGD colors in the map and, as requested, we added a PCA (please see reply 4). We have also removed the pie charts, so that sampling points are more clearly visualized (pie charts are now in Fig. S2.3). As we recognize in the main text and Fig. 3 legend, our PGD are not a perfect match to the empirical population differentiation within this taxon, but we demonstrate that their use maximizes the representation of genetic differentiation in the spatial analysis of Fig. 5, when compared to considering all the taxon distribution as a single genetic group. For better clarification we now include a bar plot (Fig. 3d) to show how each PGD is represented in the Zonation solution considering 20% of Mexico, for the "SDM" and "SDM*PGD" scenarios. As shown in the barplot, most of the spatial extent of the SDM scenario (traditional approach for systematic conservation planning) focuses on few genetic clusters included in the Western (PGD 36 and 5), Center (PGD 37) and Eastern (PGD 41) distribution of the taxon. While the "SDM*PGD" scenario increases the representation of other areas with populations likely differentiated, like the clusters represented in PGDs 48, 10, 11, 8 and 41.

Also, the admixture plots show quite a lot of admixture, with patterns that could almost be isolation by distance, which could make the admixture results somewhat suspect. I recommend the authors complement the admixture analysis with another demographic modeling approach, for example a PCA-based (e.g., PCAdapt, <https://bcm-uga.github.io/pcadapt/articles/pcadapt.html>) or maximum-likelihood based approach (e.g., Treemix, <https://bitbucket.org/nygcresearch/treemix/wiki/Home>).

Reply 4: Thank you for this suggestion. We now include in Fig.3c a score plot of a PCA (calculated with PCA-adapt, as suggested) projecting the first three components. This confirms that there is a longitudinal gradient of genetic differentiation, but also that some populations form independent clusters. Thus, to maximize the representation of genetic diversity within this taxon, more areas distributed across its range are needed, as done in the "SDM*PGD" scenario instead of larger but more localized areas as done in the "SDM" scenario (Fig. 3d). This is now also discussed in lines 234-243.

Specific comments:

Line 86: Is biodiversity actually persisting in mosaic landscapes? Consider rephrasing to emphasize instead that biodiversity is challenged by mosaic landscapes.

Reply 5: We have edited this phrase as suggested, but major edits were made to the introduction to follow other recommendations (lines 101-113, 118-119, 136-169).

Lines 87-88: second sentence not necessarily a good flow -- consider removing (it's essentially reiterated in line 92).

Reply 6: We reformulated the introduction to improve the flow of the opening argument and to emphasize the importance of crop wild relatives from a conservation perspective (lines 101-114).

Lines 92-93: not sure why crop wild relatives are particularly urgent from this sentence, consider stating more explicitly why these are important for food security.

Reply 7: We made major edits to the introduction and clarified the importance of CWR, and why it is urgent to use systematic conservation planning analyses, that consider maximizing the conservation of genetic diversity (lines 101-113).

Lines 105-110: Is the main risk gene flow from crops -> wild relatives, or is there a risk (or perceived risk by growers) of wild relatives -> crops?

Reply 8: There is also risk of gene flow in the opposite direction. We now include a line and references in that respect (lines 136-138).

Lines 111-112: Need to explain the planning based on the Magos Brehm methodology -- this is not a commonly known method.

Reply 9: We now use the term 'conservation planning assessment framework' instead of 'Magos Brehm methodology' in order to use a more familiar term (line 157). In the Methods section (lines 446-458) we briefly described the steps of this framework that we followed. Methods associated with these steps are detailed in the Supplementary Information 1 & 3.

Lines 120-124: The authors could more clearly state which aspects of the framework are novel, as they also are relying on the so-called Magos Brehm framework.

Reply 10: Thanks for the observation. We have clarified in which way our approach is novel to explicitly account for intraspecific variation (i.e. the assessment of proxies of genetic differentiation based on both historical and environmental drivers of genetic diversity) and how it fits in the conservation planning assessment framework for CWR proposed by 'Magos Brehm *et al.* 2019 (lines 148-152, 161-169, 198-211). Additionally, we formulated a definition of PGD in the main text (lines 164-169). See also Reply 13.

Lines 131-136: I recommend referring to the supplementary methods in here somewhere to direct the reader to a more detailed description. Also, consider re-phrasing to use less jargon-filled terminology, for example replacing 'CWR checklist' with 'creating a list of focal crop wild relative species'.

Reply 11: As suggested by the reviewer, we referred to the Supplementary Information and added a brief explanation on the meaning of CWR checklist, i.e. creating a list of CWR taxa distributed in an area (lines 156-163).

Lines 132-133: The focus of the manuscript and analysis seems to be on *in situ* conservation, rather than *ex situ* conservation -- but perhaps I have misunderstood.

Reply 12: The main focus of the paper is on *in situ* conservation. Still, the overall results, particularly the spatial analysis, can be used for both *in situ* and *ex situ* conservation. This is now clarified (line 160, 377-380).

Lines 186-190: I'm not sure I completely understand what 76% of proxies for genetic diversity mean -- the figure makes it seem as though this might be a proportion of a geographic distribution, but that is not how I would have interpreted a proxy of genetic diversity. I would have expected a proxy of genetic diversity to refer to some level of

within-taxon variation or divergence. I recommend that the authors add more detail to the description of these metrics and results.

Reply 13: Thanks for pointing out this was not clear. Although “genetic diversity” refers to the diversity within a taxon, more specifically the term can be interpreted in different ways, for instance nucleotide diversity, heterozygosity, population differentiation, among others. In this case, our proxies aim to identify geographic areas within the distribution of a taxon where, given the local environmental conditions and/or the isolation history of the area, interbreeding individuals within such an area are expected to show some level of genetic differentiation from individuals of other areas. To make this clearer, “PGD” now stands for proxies of genetic differentiation (now explained in lines 164-169). Also, we now better clarify the use of PGD in conservation scenarios before the results are presented (lines 260-264, 273-274).

Thus, the proxies of genetic differentiation, as we now call them, are a surrogate or substitute to represent the potential genetic differentiation within the distribution of a given taxon, which is in turn a form of genetic diversity. Because our PGD are spatially explicit, this allowed us to estimate the level of within-taxon variation based on the number of PGD covering a given SDM. For example, the range of *Zea mays* subsp. *parviglumis* is covered by 29 PGD, including 12 PGD covering most of the area and a few pixels of another 17 PGD. Of the total 29 PGD, we have SNPs data of sampling localities falling in 13 PGD (Fig. 3).

As PGD have a spatially explicit expression, we can not only estimate the number of approximate genetic clusters but also derive a measure in terms of area, i.e. estimating the representation of PGD as a proportion of a geographic distribution (Fig. 4). This is particularly relevant in systematic conservation planning in order to assess the effectiveness of a solution to cover the species-specific variation. A higher area proportion of PGD indicates a higher representation of its spatial expression. Thus, ‘76% of the area of each PGD per taxon’ is a quantitative way to express the intraspecific variation as given by the PGD covering a certain distribution range. To make this point clearer, we now added a bar plot to Fig. 3, comparing the proportion of each PGD within the distribution of *Zea m.* subsp. *parviglumis* under the “SDM” and the “SDM*PGD” scenarios. Figure 4 summarizes this same data for each taxon.

Line 200 and line 216: I was unfamiliar with the Aichi Target 13 of CBD -- consider using the entire name (Convention on Biological Diversity) and providing a link/reference.

Reply 14: We now provide the entire name. Also, we provided links to the Aichi targets 13 and 11, in particular.

Lines 199-207: This context seems more appropriate for the discussion than the results.

Reply 15: We now included this part in the discussion (lines 309-415).

Line 217: I wanted a bit more detail on what the authors meant by ‘considering 20% of Mexico’ -- did this 20% include already-protected areas? How were the 20% of terrestrial regions chosen? What was maximized?

Reply 16: We used the hierarchical map (Fig. 5a, see Fig. S2.6 for all three scenarios) a ranking of conservation priority over the entire landscape of Mexico based on the representation and complementarity of conservation features and their weighting. According

to the Zonation output, the most valuable 5% of the landscape is within the most valuable 10%; the most valuable 10% is within the most valuable 20%, and so on. This is now explained in Fig. 5 legend, main text (lines 291-294), and in the Methods (lines 524-528).

We selected the top 20% area, and used it to describe the results of the systematic conservation planning process. This area is of high conservation value, because all taxa, and on average, 50% of the area of each PGD associated with each taxon, were represented. Investing in conservation in these areas covering 20% of the territory would be particularly efficient, albeit conservation funding may be a limiting factor. Nevertheless, the selection of areas can differ based on different conservation goals, e.g. 30% as established in the CBD post-2020 strategy.

The existing network of protected areas was not considered as a criterion, as already stated in Methods (lines 521-534) and the Supplementary Information 1 (Methods, (iv) Spatial analysis, Systematic conservation planning analysis, page 7). Regarding other aspects, the use of protected areas (PA) or different criteria in conservation planning approaches could lead to a loss of efficiency to represent conservation features in the solution. The mathematical algorithms generally used for systematic conservation planning were designed to satisfy multiple objectives, and might therefore balance trade-offs among different objectives. For example, when using Zonation, PA will be fixed as areas of highest conservation value, even if this might not be true based on the distribution of conservation features, and thus focal biodiversity may not be adequately represented and protected in these areas. Thus, including PA in Zonation forces the PA areas into the top fraction of the solution, which is highly recommended when the aim is to expand on the current conservation network, but was not useful in our case, as CWR are not properly evaluated or protected in PA, and the aim was to represent conservation features efficiently in the territory. Therefore, in the case of the systematic conservation planning of CWR in Mexico, we did not include the layer into the analysis.

Lines 248-262: To me, this seemed like it belonged more in the discussion than in the results.

Reply 17: Our MS follows the Author guidelines of the journal. Accordingly, the results and discussion are combined in the Results section, while the Discussion section summarizes the main conclusions in a few short paragraphs.

Lines 278-303: This section also seemed to belong in the discussion rather than the results.

Reply 18: As stated above, our MS follows the Author guidelines of the journal.

Fig. 1: the text in the figure is a bit too small. Also, I got confused by the spatial representation of phylogeographic patterns split into biogeographic provinces, edaphology, and watersheds -- from reading the methods, I thought the phylogeographic patterns came from a literature review of genetics-based phylogeography. The methods for this step need to be more clearly conveyed.

Reply 19: Thanks for highlighting this figure needed improvement. We changed Fig. 1 in order to be more clearer on the methods and how proxies of genetic differentiation have been identified; we modified the format and reduced the number of words to increase the size of the text in the figure. We expanded on the explanation in the main text (lines 212-229) and created Supporting Information 3, where we described the methods used to

subdivide the life zones into PGD, and how each life zone was subdivided, i.e. with which spatial layer.

Fig. 5a: Unclear what the continuous map in (a) is showing -- what is 'accumulated area'? Please provide more detail in the caption, and consider a more descriptive label in the figure.

Reply 20: We now used the term 'hierarchical' map instead of 'continuous' map in order to clarify that the map represents a ranking of conservation priority over the entire landscape maximizing the representation of conservation features and considering other criteria, such as species weighting and taxa habitat preferences. This is now explained and defined in lines 297-299. According to the map, the most valuable 10% of the landscape is within the most valuable 20%; the most valuable 20% is within the most valuable 40%, and so on. Therefore, we used the term 'accumulated area'. We included a brief explanation in the figure caption to aid the interpretation of the map.
(See reply 16 for more details.)

Fig. 5b: This is said to be considering 20% of Mexico's terrestrial area -- but considering it for what? The caption should be more explicit about what is being presented. It's somewhat unclear whether the brown areas are overlaid over some of the green and brown areas (or whether they are simply neighboring each other), which would be important to know (i.e., whether the conservation areas are already in protected areas). The authors also might consider including highly urban areas on the map as well.

Reply 21: Done, we rephrased figure 5b caption to be clearer on the meaning of the 20% area and adjusted the color and transparency of layers, as suggested by the reviewer. As already stated (please see Replies 16 and 20), protected areas and indigenous were not included as criteria in the conservation analysis in order to obtain the most efficient solution to represent conservation features. Still, we thought it is important to report and show the overlap between the selected conservation area (top 20% of Mexico) and the terrestrial protected area system and indigenous areas to promote possible synergies between sectoral programs (lines 312-328).

Fig. 6: What does it mean to have the proportion of PDG area? Consider adding more context in the caption to help with interpretation of these curves.

Reply 22: PGD have a spatially explicit expression, thus it was possible to estimate the representation of PGD within the potential distribution models of each taxon in terms of area, represented on the y-axis. Also, we provided more details in the figure caption so the reader can easily understand Figure 6 without necessarily reading the corresponding document section. (See reply 13 for more details.)

Supplementary Methods

In the methods overview, please refer to the sub-sections that follow to indicate to the reader that more detail is available below.

Reply 23: Done.

Note: the plural of genus is genera.

Reply 24: Thank you for the correction. We changed it when necessary.

CWR checklist section:

The first set of criteria are unclear on what level they were applied to -- presumably not species level, since they are followed by criteria 4 and 5 for species-level. Please clarify this.

Reply 25: We have clarified how criteria were applied first at the genus and then at the species level. Please see Supplementary Information 1, Method section, (i) CWR checklist and (ii) CWR inventory, page 2.

What does it mean for a species to be included in primary, secondary, or tertiary gene pools? This terminology is not clear, please elaborate.

Reply 26: We have now included a brief description and corresponding references. Please see Supplementary Information 1, Method section, (i) CWR checklist and (ii) CWR inventory, page 2.

These methods describe each country compiling a list, but the previous section stated that the analysis is focused on Mexico -- which is it?

Reply 27: We have clarified what part of the methods and analyses are only focused on Mexico.

The 224 taxa in the final inventory are all related to the crop species, but do not include the crop species themselves -- is that correct? The wording is a bit unclear.

Reply 28: We have better clarify this point; CWR of the primary gene pool consists of wild plants of the same species as the crop.

Species distribution models:

How did the AUC test used to inform model selection? Did it always agree with AIC, or did the AUC test impact the model that was chosen?

Reply 29: We only had three taxa whose models had AIC values close to 0, but model performance, i.e. AUC, was low (0.55, 0.56, and 0.74); only one had an omission rate greater than 60%. We decided not to use those SDM.

There were some other models with AUC values ranging between 0.7 - 0.75 and omission rates greater than 20%; we included them in our analysis. We decided this based on two reasons: SDM were reviewed by experts (see reply 30) with a vast knowledge and experience in different aspects of the biology and ecology of the studied taxa, and because we didn't want to lose information about six widely distributed taxa. We marked these models in Table S2.7.

"Experts selected the best fitting SDM" -- which experts? I would expect the experts to be part of the research team writing this paper, was that not the case? How did they validate the SDMs?

Reply 30: Thank you for your observation. The experts that helped us to select between two binarized SDM were some of the ones that participated in the project's workshops, and also have authorship of the maps published in CONABIO's geoportal. All experts who

collaborated during the Darwin Initiative project have already been acknowledged in the corresponding section of this paper and are co-authors of the paper "Extinction risk of Mesoamerican crop wild relatives". Notwithstanding, the specialists, who revised the models and consistently contributed to this work far beyond the workshops, have now been invited as co-authors of this paper.

The SDM validation process was already described in the Species distribution modeling section of the Supplementary Information 1 (pages 3-4), and it is now briefly mentioned in lines 183-186.

Did using 116 SDMs mean that the number of species used was reduced from 224 to the 116 with SDMs?

Reply 31: We obtained SDM for 116 taxa out of 224 taxa, as for these taxa we had more than 20 unique occurrence data in a 1 km² grid covering the study extent. Although it was not possible to obtain SDM for all taxa, Zonation allows the inclusion of occurrence records (coordinates). We now explain how the incorporation of these data was possible in Supplementary Information 1 (Method section, (iv) Spatial analysis, Systematic conservation planning analysis, page 6) and mention it in the Methods of the main MS (lines 521-526).

Proxies of genetic diversity:

I have a few questions about the literature review that was used to characterize spatially explicit phylogeographic patterns:

- Did this review focus only on species with similar dispersal mechanisms to the crop wild relatives studied (e.g., I would expect lake fish to be structured very differently from wind-dispersed plants)?

Reply 32: Thanks for pointing out that this type of information was needed. As we now explain (lines 222-227), we targeted to represent general trends that would likely hold across species, instead of trying to represent fine idiosyncratic patterns of genetic differentiation. The patterns we used to delimit PGD cover relatively large areas (e.g. the main mountain ranges of Mexico), and hold for taxa of different dispersal abilities. For instance, populations to one side and the other of the Isthmus of Tehuantepec have been found to be differentiated in taxa as varied as plants, birds and rodents (Ornelas *et al.* 2013). Nevertheless, we focused on terrestrial species, since indeed species inhabiting lakes and rivers have different dispersal constraints. Aquatic (coastal) species were only included to subdivide a life zone covering the coasts of California Peninsula, where there are no studies on terrestrial taxa. Supporting Information 3 now includes a table summarizing the type taxa used to delimit the PGD. See replies 33, 38 for further details on taxa used in the literature review.

Ornelas, J. F. *et al.* Comparative phylogeographic analyses illustrate the complex evolutionary history of threatened cloud forests of Northern Mesoamerica. PLoS ONE 8, e56283 (2013).

- How much variation was there among the species? Did they all show similar phylogeographic patterns, or was it very species-specific? This seems critically important to assess how well the genetic diversity proxies might work.

Reply 33: Thanks for this suggestion, the variation in phylogeographic patterns is now explained in Supporting Information 3. Briefly, complete phylogeographic congruence among different taxa is uncommon, therefore we targeted to represent general trends that would likely hold across species, instead of trying to represent fine idiosyncratic patterns of genetic differentiation. For instance, although distribution ranges of highland taxa shifted during the Pleistocene climate fluctuations, in general populations persisted (glacial-interglacial periods) within the main mountain ranges, while lowland populations were ephemeral (only glacial periods), so gene flow among mountain ranges was more limited than within them. As a result, genetic differentiation among mountain ranges of different biogeographic provinces has been widely documented (Mastretta-Yanes *et al.* 2015). We used this general pattern to subdivide the life zones that occur in highlands. These types of patterns are particularly relevant for a country like Mexico, due to its complex topography, tropical and sub-tropical latitude and geographic features of different ages, which promote population differentiation among the Mexican main geographic features (as now stated in lines 187-189, 204-207). See replies 32, 38 for further details on taxa used in the literature review.

Mastretta-Yanes, A., Moreno-Letelier, A., Piñero, D., Jorgensen, T. H. & Emerson, B. C. Biodiversity in the Mexican highlands and the interaction of geology, geography and climate within the Trans-Mexican Volcanic Belt. *J. Biogeogr.* 42, 1586–1600 (2015).

The text states that of the 11832 GIS layers created, only 5004 were used -- how were these selected?

Reply 34: As now stated in the SI 1 text “Combining 116 SDM with 102 PGD resulted in 5,004 input layers with data, given that many taxa do not distribute in all of the 102 areas of genetic differentiation (PGD).”

Spatial conservation planning analysis:

In the paragraph describing the final conservation analyses, refer to the numbered approach from the previous paragraph that was chosen.

Reply 35: Done.

Supplementary results:

The supplementary figures could do with more descriptive figure captions.

Reply 36: We added on the captions in order to be clearer and to provide more context for the interpretation of figures.

Taxa distribution patterns:

The text states that the areas of highest richness are those with the highest sampling effort -- have the authors considered rarefaction analysis to evaluate whether sampling effort has been sufficient to estimate species richness?

Reply 37: Richness patterns and sampling effort analyses should preferably be done for each genus considering the reporting rate of each taxon. However, it is almost certain that

there is a bias in the sampling effort given that the main source of data is historical information from different Herbaria and collection events. Potential species distribution models (SDM) were therefore generated to eliminate sampling bias. Adjustments were made to the text in the Supplementary Information 1 (Methods, (iv) Spatial analysis, Species distribution modeling, pages 3-4) for better clarification on the decision to mainly use SDMs as the basis of analyses given the sampling effort bias.

Therefore, we consider that it is not necessary to perform a rarefaction analysis, since it was not the objective of this study to analyze the spatial patterns based on occurrence records and the uncertainty associated with sampling effort. It is worth mentioning that we have rephrased the wording in this section to not speak of "taxa richness", since this is not an analysis of complete monophyletic groups (i.e. not all Angiospermae; or a given family or genus). We only highlight the areas that concentrate a high number of studied taxa using a 5 km² grid of presence/absence data and that broadly seems to coincide with areas of high concentration of CWR related to the nine crops studied according to the SDM diversity pattern (referred to in the text as "high taxa richness" areas).

Reviewer #2 (Remarks to the Author):

This paper develops a novel way to incorporate within-species' genetic diversity/ evolutionary resilience into systematic conservation planning (i.e. data-driven ranking of what areas should be protected for maximizing biodiversity conservation). To my knowledge, the approach- with proxies of genetic data (PGD)- has not been used before. Although genetic data has been considered recently in a few studies, research up to now has only included at most approximately 10 species, and generally required genetic data (e.g. Hanson et al. 2021 Cons Biol). Other research has assumed that percent of species area equates to a good proxy of genetic diversity (Khoury et al 2020 PNAS). This paper is a major advance on such earlier work because it explicitly includes phylogeographic divisions (from past genetic studies), but not actually requiring genetic data. Thus the approach can be applied to any number of species at national or even global scales. The authors compare the performance of their approach to alternatives such as only using species distribution models, and calculate both the percent of species' ranges and percent of PGD conserved. They also experiment with including other information as weights (IUCN status for example) and other data including species' habitat preference and land use. I think it is a major contribution to the literature, and is highly timely for conservation policy (for example for the Convention on Biological Diversity and Sustainable Development Goals).

I have numerous requests for clarification about methods. I have other requests for clarification because the authors have evaluated many different scenarios and it is a bit hard to follow. However, I do not see any major flaws in the work.

Larger requests

Methods- line 164- How many species were included in this literature review? Were they all non CWR? Do they have similar characteristics, geographic range, etc. as crop wild relatives? In other words, are they well representing CWR genetic 'breaks'?

Reply 38: Thanks for pointing out that this type of information was needed. We now detail which type of species were used in the newly created Supporting Information 3. Briefly, to show that our approach can be used without prior information on CWR, we excluded from the literature review studies performed in CWR, but as more studies on such taxa become available, they can be used to fine-tune the PGD.

As for the characteristics and ranges of the taxa included, since most of the life zones cover large territories, and complete phylogeographic congruence among different taxa is uncommon, we targeted to represent general trends that would likely hold across species, instead of trying to represent fine idiosyncratic patterns of genetic differentiation. We followed this approach because the CWR included in this study vary considerably in their distribution ranges (from the coastal *Gossypium* to the cloud forests *Persea*, and from micro endemic to widely distributed), so instead of focusing on the CWR distribution, we aimed to subdivide each life zone in the patterns that more likely would hold across any taxa distributed within them. See replies 32 and 33 for further details on species used in the review.

Lastly, please notice that we changed the wording and now use "proxies of genetic differentiation" instead of "proxies of genetic diversity". See reply 13 for details.

Methods/ Results- line 164- It is not clear to me if phylogeographic breakpoints were actually used. It looks like in Fig 1 that divisions such as watersheds and soil were used. But was there any method used to actually include breakpoints indicated by genetic data (attempts have been made in N America for this, e.g. Soltis et al 2006, Shafer et al 2010)? If not, the text should be more clear that phylogeographic genetic datasets were not used explicitly but rather generally support breakpoints at watersheds etc.

Reply 39: To translate the phylogeographic information into a spatial context, we used biogeographic regions, basins, topographic or edaphic data to split the life zones into different subzones using the best fitting cartography to represent the phylogeographic patterns. In Supporting Information 3, we now detailed how each life zone was subdivided, i.e. with which spatial layer. We know it is not trivial translating the genetic data into a spatially explicit feature. Still, we considered the polygons we used are a more objective and reproducible way of subdividing a life zone than using a "straight" line as often phylogeographic breaks are shown in reviews and meta-analysis.

Methods/ Results- line 186 onward- Even after reading the Supplemental its hard to picture the difference between PGD+SDM and PGD*SDM. Please explain more. Also, why not include a PGD only scenario (just PGD).

Reply 40: We rephrased the main text (lines 259-264, 273-274) in order to be clearer on the difference between both scenarios ("SDM+PGD" and "SDM*PGD"), which differ mainly in the number of conservation features and whether proxy of genetic differentiation (PDG) were used independently, as in the "SDM+PGD" scenario, or as layers to divide each taxon distribution to identify potential genetic variation across their geographical range, resulting in 5,004 conservation features, as used in the "SDM*PGD" scenario. This is also detailed in the Methods section now (lines 502-513). Also, we added a Fig. S2.5 to provide an overview of the spatial analysis and its inputs.

We consider that using only PGD as a conservation feature without the information of CWR taxa does not make sense for assessing conservation areas for species or any other biological group. It would most likely make a poor surrogate to represent biodiversity features, as has been demonstrated in many systematic conservation planning studies that used environmental surrogates, such as soil or land types.

Methods/ Results- line 213 onward- I'm not sure how taxon specific habitat preferences are incorporated. Are some habitats 'masked out' or removed from the SDM? Or down weighted pixels?

Reply 41: We expanded on the description on how habitat information was included in the analysis in the Supplementary Information 1 (Method section, (iv) Spatial analysis, Systematic conservation planning analysis, pages 5, 7). We used the information to refine the conservation areas by assigning specific weights to high, low or no habitat preferences, thus down-weighting the pixels with low preference of occurrence for a given taxon.

It is also not clear how occurrence records are incorporated into the Zonation procedure? It would also be great to read a few sentences on how many occurrence records are sufficient for a species.

Reply 42: We now included a more detailed description of how occurrence records were considered in the Supplementary Information 1 (Method section, (iv) Spatial analysis, Habitat

preference (page 5) & Systematic conservation planning analysis (page 7). There is no minimum or maximum limit of the number of taxa and coordinates that can be included in the Zonation software using so-called “SSI files”

For obtaining SDM, we assessed that there must be more than 20 unique occurrence data in a 1 km² grid covering the study extent. In our case, one taxon was either represented by a SDM or its occurrence records (see Table S2.3). (Please see reply 31 for more details.)

Results- It looks like the preliminary analysis resulted in a max of 76% of PGD per taxa (line 186) while the final analysis (including IUCN ranking etc) resulted in only 50% of PGD per taxa (line 237). Why the major drop?

Reply 43: The final analysis considered the information on habitat among other criteria, which was not included in the preliminary analyses. This directed the selection to areas that more probably contain CWR, and masked out areas that have already been converted to land uses not preferred by a taxon or not ideal for CWR persistence. As already stated in the text *“for many taxa a considerable amount of their habitat has already been lost, degraded and fragmented within their potential range”*.

We included an additional figure in the Supplementary Information 2 (Fig. S2.5) to provide an overview of the different scenarios and their input data, as suggested by the reviewer in the following comment.

Higher level- because there are so many scenarios analyzed, it can get confusing to follow the methods- a flow chart listing them all would be useful- the main Fig 1 only looks at the preliminary scenarios. A flowchart showing what goes into each scenario would be great- and matching that to Figures. Could be Supplemental.

Reply 44: Thank you for the observation. We now included an additional figure in the Supplementary Information 2 to provide an overview of the different scenarios and their input data (Fig. S2.5). Also, we changed Fig. 1 to provide a general workflow of the project, highlighting the development and use of the PGD for the systematic conservation planning assessment.

A higher level discussion point/ critique. I think that selection of areas using Zonation will not consider connectivity, e.g. cells selected are at 1 square km (or whichever resolution) which could lead to selection of very tiny parcels of land which are isolated. Could you discuss this problem briefly? Such a prioritization would also not consider if the habitat (for example one square km) is large enough to support a viable population (such as effective size 500), which is also an indicator of genetic diversity. This might mean that genetic diversity is not sufficiently conserved!

Reply 45: We strongly agree on the importance of connectivity and connected area networks. For the present study, we discussed different approaches to consider habitat connectivity using Zonation. Nonetheless, we finally decided to run the analysis without particularly accounting for connectivity based on conceptual and technical aspects. Limiting the number of isolated pixels generally signifies trade-offs, that is reducing the efficiency of the solution to represent taxa and including lower-quality habitats. Still, throughout the text, we emphasized the importance of conserving ecological and evolutionary processes, which necessarily implies the connectivity of areas and large enough effective population sizes. This is now briefly discussed in lines 311-350, and 394-398.

Moreover, the conservation area proposal (Fig. 5, Figs. S2.6 and S2.11) showed an observable aggregation of conservation areas, as already described in the main text. To provide a more detailed response on the aggregation pattern, we evaluated the 20% conservation area proposal considering all taxa (Fig. 5b). There were 30,725 planning unit clusters of which 4,458 had an extension larger than 5 km, covering 90% of the 20% conservation area. The remaining 26,267 clusters were equal or less than 5 km, accounting for only 10% of the conservation area, including 16,898 isolated pixels of 1 km², representing 3,6% of the conservation area. The aggregation pattern might be due to overlapping taxa distributions as well as taxa that have range-restricted distributions isolated from the other taxa (Fig. 2). In any case, the resulting output of the systematic conservation planning process shows the most efficient solution in terms of CWR representation in the study area (Fig. 6). It offers an important basis to guide decision-making for the conservation of CWR in Mexico, which was one of the main objectives of the study.

Regarding the pixel size or planning unit in spatial analysis, they also have a strong influence on the final conservation area network. A larger pixel size, eg. 10 x 10 km or 50 x 50 km, generally includes different habitats given the considering heterogeneity of the territory and also including non-viable habitat, but more importantly they might also include areas where no target taxa or only a few widespread common, non-threatened taxa occur, which could be conserved more efficiently in other sites. In contrast, smaller cell sizes avoid this kind of inaccuracy, thus leading to more realistic solutions, and provide the opportunity for local and tailored strategies for effective conservation (as mentioned now in lines 422-423). Also, smaller cell sizes have been recommended for heterogeneous landscapes, including a mosaic of agricultural, urban and forested areas (Mo *et al.* 2019, available at <https://www.google.com/url?q=https://www.sciencedirect.com/science/article/pii/S235198941930023X&sa=D&source=docs&ust=1635286925760000&usg=AOvVaw14PQaP3qy0DFZTcnQMmlku>), as is the case in Mexico.

In order to ensure that genetic diversity is sufficiently represented, we developed the PGD and evaluated the representation of conservation features considering different scenarios (Figs. 3 and 4). Based on the “SDM*PGD” scenario, we selected the top 20% area as this area is of high conservation value, because all taxa, and on average, 50% of the area of each PGD associated with each taxon was represented (Fig. 6).

It is worth mentioning that recently, a GEF funded project (<https://www.thegef.org/project/strengthening-management-effectiveness-and-resilience-protected-areas-safeguard-biodiversity>) executed by the National Commission of Protected Areas, Conabio (2020) identified bioclimatic corridors that connect remaining fragments of primary vegetation in Mexico along climate gradients, avoiding areas of high human impact. These corridors might facilitate the mobility and dispersal of individuals of various species across habitats in the face of global climate change (<https://www.biodiversidad.gob.mx/pais/planeacion-para-la-conservacion/corredores-bioclimaticos>). To complement this reply, we evaluated the proportion of the 20% conservation proposal area for CWR of the scenario that considered all taxa (Fig. 5b) and its overlap with the bioclimatic corridors (please, see the figure below). Twenty one percent of the conservation area of CWR are also important to connect vegetation fragments, and should be recommended for sustainable management actions, such as agroecological practices and ecological restoration actions that might enhance connectivity between habitats.

A suggested analysis, if it's possible- how many taxa have their PGD conserved at very low levels, say less than 60%? It looks from Fig 4 that very few... but which are they, and why? In other words, why are some taxa PGD conserved very low- are they all in the same genus, spatial location, and have similar biology? Is there anything in common to them? Also how many have >90% PGD conserved? The current draft of the CBD post 2020 framework (released today) suggests 90%.

Reply 46: Thank you for suggesting this analysis, we carried it out. Just a small clarification, Fig. 4 does not show the proportion of PGD *conserved*, but *represented* in the Zonation solution.

The taxa with less than 60% of the mean area of their PGDs represented in the Zonation solution are:

Mean prop. of PGD	Total Area of the SDM (km ²)	Threat category	Taxon
0.3748263	11810	DD	Cucurbita cordata
0.4883239	32037	LC	Cucurbita digitata
0.5813091	12183	LC	Cucurbita lundelliana
0.5175327	13114	DD	Cucurbita palmata

0.3760386	17106	VU	Gossypium davidsonii
0.5506088	53446	NT	Persea podadenia
0.5892198	30141	LC	Phaseolus angustissimus
0.531142	31513	LC	Phaseolus filiformis
0.5789563	144453	LC	Phaseolus lunatus silvester
0.5429136	87717	<NA>	Physalis acolia
0.5492944	277396	LC	Physalis angulata
0.5622299	46563	LC	Physalis crassifolia
0.5918319	316674	LC	Physalis hederifolia
0.5959233	328147	LC	Physalis patula
0.5527966	305552	LC	Physalis pruinosa
0.5600602	401532	LC	Physalis pubescens
0.5835505	71380	LC	Tripsacum lanceolatum

The taxa with more than 90% of the mean area of their PGDs represented in the Zonation solution are:

Mean prop. PGD	Total Area of the SDM (km ²)	Threat category	Taxon
0.911952	21704	EN	Capsicum lanceolatum
0.9348443	12008	VU	Gossypium gossypioides
0.9530186	14472	EN	Persea albida
0.9010968	8179	LC	Persea caerulea
0.9115334	10910	VU	Persea donnell smithii
0.9036658	27794	EN	Persea pallescens
0.9131448	9487	LC	Persea vesticula
0.918431	22707	LC	Phaseolus jaliscanus
1	341	LC	Physalis ignota
0.9397357	4969	LC	Solanum agrimonifolium
1	142	VU	Solanum clarum
0.9384119	24517	LC	Solanum hougasii
0.952277	10185	NT	Solanum trifidum
0.9728131	4788	EN	Zea diploperennis

There is no pattern at the genus level on how PGDs are represented but, as expected, Zonation favors the representation of threatened species (panel (a) in figure below). So these taxa tend to have a higher proportion of their range represented in the solution considering 20% of Mexico, as well as more than 85% of the mean proportion of PGD represented.

For widely distributed taxa (light blue colors in panel (b) in figure below) it is not possible to represent all their range in 20% of Mexico, so they tend to be less represented both in terms of their distribution range and PGD.

We now mention these findings in the main text (lines 266-270). However, we did not include the results of this analysis in the ms, because the results are somehow expected, and as stated by the reviewers, the ms is already complex enough. However, we can include it in supplementary materials if required.

Smaller requests

Line 112 briefly define what you mean by "Magos Brehm et al methodology"

Reply 47: We rephrased and clarified that the methodology refers to a conservation planning assessment (line 157) and provided more information in the Supplementary Information 1 (Methods, page 1). This methodology was designed to develop and implement national conservation strategies and action plans of CWR. The approach has been used at least partly by countries and regions like the UK, Norway, Turkey, Czech Republic, South Africa, China, South Africa, Argentina and South America, among others. We now omit the use of the term 'Magos Brehm methodology' as it may not be known by the reader and in order to be more specific. See also reply 10.

Methods/ Results- SDM- what probability of occurrence was used as a threshold to (from Supp Matt) "trimmed continuous SDM using binary SDM"?

Reply 48: We rephrased and clarified this in the Supplementary Information 1 where we described the SDM modeling process (page 3-4, 9). We used the ten percentile and the minimum training presence threshold to binarize continuous SDM, and with the help of experts selected the model that more appropriately represents current species distribution. Most models were binarized using the ten percentile; eight models were binarized using the minimum training presence threshold, as indicated in Table S2.7.

Methods/ Results- line 175- is there any way to better measure "no complete coincidence" between genetic and PGD data for maize? It looks like it was only compared "by eye"? Also is this a crop wild relative or an actual crop? Could other datasets be used for validation (e.g. those with chloroplast or microsat data)?

Reply 49: We have modified Fig. 3 to better show how the empirical genetic differentiation relates to our PGD. We hope the new color bar below the admixture analyses allows to better compare the empirical clustering against the PGD subdivision of the species range. To demonstrate that their use maximizes the representation of genetic differentiation in the spatial analysis of Fig. 5, we now include a bar plot (Fig. 3d) to show how each PGD is represented in the Zonation solution considering 20% of Mexico, for the "SDM" and "SDM*PGD" scenarios. See replies 3 and 4 for further details and discussion. We also now clarify that *Zea mays* subsp. *parviglumis* is a CWR of maize (*Zea mays* subsp. *mays*) (main text lines 231-232 and Fig. 3 legend). Chloroplast or microsat data could also be used to validate our approach, but we chose this SNPs dataset because it includes both neutral and non-neutral loci, as well as finer resolution than the above mentioned markers. We also now better explain that as more data is available for CWR species, PGDs can be further corroborated and fine-tuned (lines 252-254).

Figure 6 has PDG instead of PGD on y axis

Reply 50: Thanks for the observation, we corrected it.

Methods/ Results- line 272- Supp fig 8... these panels do not look very different to me- I think they look very similar. Can you point out some differences?

Reply 51: We agree the representation curves show similar patterns, still there is one evident difference which we now described in Supplementary Information 1(Results, Conservation areas, page 10): "A major difference is evident regarding the taxa that are critically endangered; most of them distributed in different habitat types, so their representation was favored in the corresponding scenarios, i.e. (a) and (c), but not in scenario (b) that focused on taxa exclusively distributed in natural vegetation."

Discussion line 316- "this can inform public policy regarding living modified organisms such as crops and agriculture subsidies in order to mitigate threat processes to CWR." To me this seems like a bit of a stretch, or maybe is just not clear to me. Can you explain?

Reply 52: We argue that this type of approach can inform decision making and public policy formulation, particularly risk assessments and public policies related to the release of living modified organisms (LMOs). The latter must be understood in the context of the presence of a rich genetic diversity related to LMOs, which is critical for example for maize or cotton in Mexico (Acevedo *et al.* 2016, available at https://link.springer.com/chapter/10.1007%2F978-1-4614-6669-7_21). Furthermore, our results can be related to where and when agricultural subsidies can or not affect the genetic diversity of CWR (see Alimentar a México sin Deforestar, available at <https://bioteca.biodiversidad.gob.mx/janium-bin/detalle.pl?id=20210831165727>).

Reviewer #3 (Remarks to the Author):

This article provides a thorough analysis to identify priority areas for in situ conservation of crop wild relatives in Mexico. It also provides a methodological approach to conduct such prioritization that incorporates evolutionary and threat assessment information. The article is the result of a large international multidisciplinary collaborative project and contains very relevant data and analyses related to crop wild relative conservation in Mexico. The results, i.e. the identification of priority areas for in situ conservation of CWR in Mexico, are noteworthy, especially given the fact that Mexico contains some of the most relevant hotspots of CWR diversity in the world. These results certainly add great information to the current knowledge of CWR diversity in Mexico.

Having indicated this, the manuscript as it stands, is, in my opinion, at an early immature stage and has deficiencies concerning its purpose, focus and delimitation. Both the title and the last paragraph of the introduction suggest that the purpose of the manuscript is to present a new methodological approach for crop wild relative conservation that includes evolutionary and threat assessment information in the process.

However, there is nothing of great relevance in this methodological approach. As indicated at the beginning of the Results section it basically follows the main steps of Magos Brehm et al (2017, 2019) proposals.

Reply 53: We made major edits throughout the document (see lines 148-169, 281-285, 466-534 and replies 9, 10, and 47 for more details and line numbers where changes were made) and Supplementary Information 1 (pages 1, 2, 5, 6, 7, 9), and included Supplementary Information 3 to highlight the objectives and novelty of our approach to incorporate within-species' genetic diversity into systematic conservation planning assessments in the absence of genomic information, through the use of PGD, which to our knowledge have not been used before. Also, we more clearly emphasized the differences to previous CWR systematic conservation planning exercises that apply the minimum set cover problem; our approach maximizes the coverage of range-wide genetic diversity in conservation network proposals (lines 141-147). Please note, we changed the wording and now use "proxies of genetic differentiation" instead of "proxies of genetic diversity" (see reply 13 for details).

The proposal does present a significant addition which involves the use of phylogeographic studies information as a proxy of the demographic history of the target CWR species. This way, in the classification of the territory that is going to be used as a proxy of within-species genetic diversity, in addition to the use of Holdridge life zones to account for differential environmental conditions, the authors use a synthesis of phylogeographic studies to account for the genetic diversity among populations that may have been generated through the past history of the species. Surprisingly, if this is the novelty of the study, and the purpose of the article is to present it as such, the manuscript provides no further information on how this is done other than "Then each life zone was divided according to phylogeographic patterns of non CWR species, using biogeographic regions or topographic and edaphic data to define the cutline". The authors provide a list of 42 references of phylogeographic studies that were used to conduct this process. Concerning this part of the methodology, there is not enough detail provided in the methods for the work to be reproduced.

Reply 54: Thanks for highlighting that this part of the methodology required more detail. We now provided a more detailed description on how proxies of genetic differentiation* have been assessed so the methodology can be reproduced. For this we now include Supporting Information 3. Briefly, SI3 explains how the literature review was conducted, the rationale and cartographic data behind how each life zone was subdivided, a Figure on Mexican biogeography and main geographic features to guide the reader, and a table summarizing the type of taxa included in the review. We also extended the reference list to around 60 studies. Please see replies 19, 32, 33, 38 and 39 for more discussion on this new Supporting Information and line numbers where changes were done to the main text. We also expanded on the main differences with other CWR conservation planning approaches to clarify the novelty of the approach (lines 201-207). Please see Replies 10 and 47 for more details.

* Please notice that we changed the wording and now use “proxies of genetic differentiation” instead of “proxies of genetic diversity”. See reply 13 for details.

I think it is a good idea to try to incorporate some criteria into the classification of the territory that is going to be used as a proxy of genetic diversity that takes into account the demographic history of the target CWR. However, a) I do not think it is possible to correctly infer genetic diversity associated to the demographic history of the target CWR by extrapolating results from other life forms that include birds and reptiles, b) in any case, the authors give no information or discussion to support their claims, c) they neither provide a detailed description of how this partition of Holdridge life zones was done using the information resulting from the phylogeographic studies, d) they neither discuss how this approach can or cannot be applied to similar territories in other parts of the world.

Reply 55: It is good to read that you like the idea of classifying the territory to better represent genetic diversity. Thank you for highlighting that those important details were very much needed in order to understand our approach. We hope Supporting Information 3 and the new wording in the main text (see line numbers below) better confer our ideas now. As for your point concerns:

a) We now realize “genetic diversity” is too wide of a term to be interpreted equally by all researchers, thus, we now explicitly narrowed it to “genetic differentiation”. Thus, PGD stands now for “proxies of genetic differentiation” (see reply 13 for details). Demographic history indeed plays a role on generating population differentiation, but rather than aiming to infer the demographic history of a given CWR (e.g. whether its populations contracted or expanded since the last glacial maximum) we aim to identify geographic areas within its distribution where population differentiation would be expected relative to other areas. Please see replies 32 and 33 for a detailed discussion of our rationale and line numbers where changes were made.

b) Supporting Information 3 now briefly explains our rationale and claims, and the analyses of Fig. 3 (genetic data of *Zea mays* subsp. *parviglumis* supports it with empirical sub-genomic data). See replies 3, 4, 13, 32 and 33 for changes to the main text and further discussion on this analysis.

c) Table S3.1 of Supporting Information 3 now details exactly how each life zone was subdivided in different PGD. This includes a brief rationale, references and which cartography (biogeographic provinces, basins, topography or edaphology) was used to subdivide each of the life zones.

d) We designed our approach to be repeatable in any country. That is why we used Holdridge life zones (which can be estimated for any territory in the world) instead of other sources of data specific to Mexico. We also excluded CWR from our literature review (see reply 38) to prove that our approach can work without phylogeographic data on the target taxa. This is mentioned in lines 231-232. Finally, we now detail (lines 245-251) that even if no literature on phylogeography is available, it is still possible to assume genetic variation based on biogeographical, climatic and geological conditions that shape biodiversity and diversification patterns. Although we recognize that more empirical data is needed to evaluate this. Important areas for CWR using the PGD methodology in this way to define conservation of El Salvador are shown in a synthesis report in Spanish, available at <https://bioteca.biodiversidad.gob.mx/janium-bin/detalle.pl?Id=20210906173606>

Mastretta-Yanes, A., Moreno-Letelier, A., Piñero, D., Jorgensen, T. H. & Emerson, B. C. Biodiversity in the Mexican highlands and the interaction of geology, geography and climate within the Trans-Mexican Volcanic Belt. *J. Biogeogr.* 42, 1586–1600 (2015).

Apart from this, the contents of the Results section are fuzzy. If the purpose of the article is to provide a new methodological framework, the Results section provides little clear description of what the new framework is other than what is depicted in Figure 1. Even the description provided in its legend is minimal. If there are other novelties in the framework (e.g., the comparative analyses of target CWR taxa depending on the nature of their habitat, the use of the Zonation software) than the use of evolutionary information, they should be clearly identified, described and presented in a first presentation of the methodology.

Reply 56: Thanks for highlighting this was needed. We made major changes in the Result section (see below), expanded the Methods accordingly (lines 446-534) and improved Figure 1. For changes to Fig. 1 and general description of our workflow see reply 19. As for the Results section, first we rephrased two main paragraphs in order to highlight the conceptual framework and development of proxies of genetic differentiation (lines 197-229). Second, we clearly acknowledged different attempts to capture genetic diversity and indicated the main differences of the frameworks (lines 202-207). Third, as we ran five preliminary and three final scenarios, we acknowledge that it might be difficult to follow the methods, thus we have now included an additional figure (Fig S2.5) to provide an overview of the systematic conservation planning analysis performed in the context of the present assessment to identify important areas for conservation of Mesoamerican CWR. We also added subheadings to better organize the readability of the Results section.

On the contrary, the Results section is mainly dedicated to present a detailed account of the results of implementing this methodological framework in a selection of target CWR for Mexico. In this sense, I feel that the value of this work precisely resides in these results because as I previously mentioned, I think that the methodological innovation concerning the use of evolutionary information is not sound and it is not properly explained.

Reply 57: Thank you for recognizing the importance of the results. To emphasize the methodological innovation, we provided a more detailed description on the development of PGD, as mentioned in replies 19, 32, 33, 38, 39 and 55. Also, we included a brief description on the assessment of PGD if no information of phylogeographic patterns is available for a country or region, which was the case of El Salvador that were also part of the international project (see reply 55). We also modified our Introduction, Results and Discussion sections (lines 139-147, 198-202, 409-415, 420-425, 426-429) to emphasize the value of our methods and results for CWR conservation.

However, when it comes to this, there is an additional problem: the results of implementing steps I, II and III of the methodology have already been submitted for publication (Goettsch et al., in prep.), but they are not published. The authors provide a copy of the manuscript that has been sent to the journal *Plants People Planet*. There is no duplication of its contents, but the fact that the article has not been published makes it an unreliable reference (most journals only accept references concerning published information). Furthermore, in this line, citation number 47 (Urquiza et al) is also a reference to a document that has not yet been published.

Reply 58: We are grateful to announce the paper of Goettsch *et al.* had been published, and available at <https://doi.org/10.1002/ppp3.10225>. We omitted the reference of Urquiza *et al.* since it was not really necessary to support the steps presented in this MS.

The Species Distribution Models have been performed rigorously and are described in great detail.

Reply 59: Thank you very much for the positive comment.

One key result in the implementation of the methodology for Mexico is the generation of the proxies of genetic diversity (PGD). The authors claim partial support of their PGD by contrasting them with empirical data from the teosinte *Zea mays* subsp. *parviglumis* obtained from a doctoral thesis (Figure 3). There is certainly an association between the genetic clusters obtained through molecular analysis and the corresponding PGDs. It is a pity that the results of the doctoral thesis are not yet published in a scientific journal (or at least there is no indication of this in the manuscript) to provide the possibility of further verification or analyses.

Reply 60: We have improved Fig. 3 and its discussion to better show the association between our PGDs and population differentiation within *Zea mays* subsp. *parviglumis*. Please see replies 3, 4 and 13 for details. The results of the doctoral thesis are still being processed as part of a larger study including other genomic data. However, the author of the thesis (and also co-author of this MS) kindly agreed to first publish the data of *Zea mays* subsp. *parviglumis* as part of this study. If the data is needed for review purposes, we made it available as a "related manuscript file". Instructions on how to use the data is available in the scripts https://github.com/CONABIO/analisisUniCons_proxiGen/blob/main/bin/PCAdapt_PGD_teocintles.Rmd and https://github.com/CONABIO/analisisUniCons_proxiGen/blob/main/bin/plot_admixture_PGD_teocintles.Rmd of our already public Github repository. These and all data used to make the

analyses presented here will be included in the Dryad repository associated to this MS, once it is accepted.

Some other parts of the implementation of the methodology include some additional analysis that may be of interest if its relevance is properly explained and discussed, but otherwise they contribute to the loss of focus. This is the case of the three independent analyses performed to different sets of CWR depending on their habitat preference. In the way they are currently presented they rest clarity and focus to the presentation of the results. Are these tests relevant? Do they really contribute to the purpose of this article? What is the use of the results presented in Supplementary Table 7 and Figs 6, 7, 10 and 11 in the context of the purpose of this article? What is their practical application?

Reply 61: We agree the paper includes quite a lot of information and analysis as it represents some of the outputs of a large international collaborative project (see Fig. 1). We hope that the new Fig. 1 and associated changes in the methods and results help to clarify the usefulness of the analyses the reviewer mentions (see reply 19). The figures in question are related to three scenarios that considered (a) all taxa, (b) taxa exclusively distributed in natural vegetation, and (c) taxa associated to a wider range of habitats. They are not intended as simple tests to explore the effect of running independent analysis when using different datasets. Including habitat preferences in the assessment to identify important areas for CWR conservation allowed to go a step further in fine tuning and accurately assessing the areas proposed as for *in situ* conservation actions. This is relevant because some CWR are distributed only in primary vegetation (e.g., *Persea* in cloud forests), while others can tolerate other habitats (e.g., herbaceous taxa that prefer secondary vegetation). Also, it allowed the exploration of a set of possible conservation actions specifically targeting certain taxa given their habitat preferences. Consequently, each scenario has different and particular consequences for CWR conservation, which is fundamental to comprehensively inform and guide conservation action plans and public policy. This is mentioned in lines 359-374, 426-429). Therefore, we consider that the presented data, tables and figures are informative and important for guiding conservation action information.

This is also the case of the tests of different alternatives to incorporate PGD into the spatial analysis. It would be clearer and more focused if Figure 4 would only present the results corresponding to the alternative finally used for this study (leaving the rest, or the comparison, in a supplementary Figure).

Reply 62: Thank you for this comment. After much consideration and internal debate, we decided to leave Fig. 4 as it is, since it is a key figure that shows how output of the systematic conservation planning assessment really changes when incorporating the PGDs, and supports our decision to incorporate the PGDs into Zonation in the way we did. Having the different scenarios in the same figure also helps to better understand Fig. 3d. Finally, as highlighted by reviewer 2 in reply 46, it allows to see there is variation on how well species are represented by the different scenarios.

Another formal deficiency that needs to be improved concerns the assignment of contents to Results, Discussion and Methods. Depending on whether the article is focused as a methodological approach or as a conservation assessment of target CWR in Mexico, the Results section should be streamlined moving some of its contents to the Methods and

Discussion sections. It is noteworthy that the Results section contains considerable interpretation of the results and association to other references that would probably fit better in the Discussion section. In this sense, the current Discussion section made out of just two paragraphs connects more with the idea of some concluding remarks than a proper discussion of the results.

Reply 63: Our MS follows the Author guidelines of the journal. Accordingly, the results and discussion are combined in the Results section, while the Discussion section summarizes the main conclusions. We have however improved the Results and Methods sections, hoping that these changes make the paper easier to understand. See replies 9, 10, 13, 40, 32 47, 53-57, 61.

In conclusion, there is a need to clarify the purpose of this article. I think the results of the assessment of CWR conservation in Mexico are much more relevant than the novelty of the methodology and I would suggest that the article is reoriented in this way.

Reply 64: Thank you for highlighting it was needed to clarify the purpose of our work. We have rewritten and reorganized the document to emphasize the importance of assessing PGD in the context of CWR conservation in Mexico, for which it was necessary to propose a novel methodology that could maximize the coverage of range-wide genetic diversity in conservation networks proposals. This is now stated in the Abstract (lines 88-92), Introduction (lines 101-113, 118-119, 124-125), and Results (lines 197-211).

In any case, the contents of the manuscript must be properly developed in the corresponding Methods, Results and Discussion sections, and only the information that is necessary to support the main purpose of the article should be included, eliminating other information that is less relevant. Citations to unpublished references should be eliminated if possible and the relationship with information about other parts of the project should probably be dealt in a different way.

Reply 65: We have extensively modified the main text and key figures to better reflect our approach and main findings of the study. We also have added subheadings to the Results section to improve the readability. Please see replies 19, 56, 58, 63 and 64. We eliminated citations to the remaining unpublished reference, as the other has already been published now.

Reviewers' Comments:

Reviewer #1:

Remarks to the Author:

The authors have revised the manuscript and resolved many of my initial comments and concerns. I have noted several parts of the paper that could still use improvement, including some improvements to the figures.

I still find the percentages of proxies of genetic differentiation to have a non-intuitive and somewhat confusing definition in this paper, and I don't fully understand how the definition matches up to a percentage. Is it that across the entire range, if a PGD is 75%, that 75% of the individuals will fall into one genetic grouping and 25% into another? Or is it that 75% of the distribution area is one cluster associated with that PGD? Do proxies overlap geographically, and how is this dealt with when it comes to these proportions/percentages? I believe this needs to be much more clearly articulated for the paper to be understood by a general audience.

Specific comments

Line 165 references PGD before the abbreviation is defined (defined on line 170). My personal recommendation is to not abbreviate 'proxies of genetic differentiation' at all.

Fig. 1: I found the figure a bit difficult to follow. I wasn't sure what the colours meant, or what the papers and lines on the 'phylogeographic patterns' section were supposed to show. I also did not understand how the puzzle pieces at the bottom fit into the overall pattern. The formatting with bolded boxes and colour in the example box drew my eye away from what the actual final output is, which is the 'Systematic conservation planning' box. I might suggest removing the example, as I found it distracting from the main point of Fig 1 (an overview of the approach) and it's somewhat repetitive with Fig 3.

Lines 186 - 188: A brief mention of what data were used (occurrences + habitats is what I understood from the methods) would improve readability of this section. Also, this was not clear from Fig 1, so consider whether the flow diagrams could be improved.

Lines 195 - 197: I'm not sure I follow the comparison here. It sounds like the previous study used 35 taxa, and this study used > 200, so I'm not sure I follow the statement about using only 10% of the taxa, especially since the following sentence states that 167 taxa were estimated in a 15x15km grid, which sounds like more than the estimates reported here estimates. Furthermore, Fig. 2 shows the highest category for taxa as 31-52, so differentiating between the 35 taxa vs >50 is not possible in the graph. I suggest re-wording parts of this section for clarity, and possibly consider re-making Fig. 2 with narrower groupings if this comparison is important.

Fig. 2: I'm not sure if this figure is entirely necessary to include in the main text, especially if the authors are motivated to reduce the number of main text colour figures.

Lines 212-214: Consider signalling here that the proxies mirror genetic differentiation in at least your 1 example

Lines 234-235: The phrasing of this sentence was a bit confusing. Consider instead something along the lines of "Zea mays ... was not included in the literature review so as not to have its phylogeographic patterns influence the estimation of proxies for genetic diversity for this species. This approach allowed us to mimic the scenario presented by the many species without any existing genomic resources."

Lines 237 - 241 and Fig. 3: The figure and the text here seem to show that there is some isolation by distance, which makes the inferences of clusters difficult to determine. How much does this impact your interpretations?

Lines 241 - 246: This section was very confusing to me as written. In what way did the approach increase representation of areas? Maybe point to the specific part of the figure that shows this (Fig. 3d, I think). I was unclear what "there is no complete coincidence between the PGD and the

empirical genetic data" actually means, or where this is shown.

Fig. 3: I found several parts of this figure to be difficult to follow. 3a and 3b are reasonably straightforward, although some of the colours are difficult to tell apart (the two yellows in the admixture plot, the two brownish greens in the PGD topologies). But the points were very small and the colours almost impossible to tell apart in Fig 3c, and the legend should specify that this was a PCA of the genetic data. For 3d, I was confused about what the proportion is and how it can be assigned to each proxy of genetic differentiation, especially in models without the proxies for genetic differentiation. Should the proportions sum to 1?

To me, it seems like the main goal of Fig. 3/the analysis with *Zea mays* is to demonstrate that the proxies of genetic differentiation are decent proxies, and I'm not entirely convinced.

A secondary goal of Fig. 3 is to demonstrate that the SDM + PGD is better than just the SDM, but I think Fig. 4 does a better job of showing this, so I would suggest removing 3d from Fig 3 as it detracts from the primary goal of the figure.

Fig 4 and lines 260 - 275: I find the use of the proportion of taxa distributions and proportion of PGD areas confusing metrics of performance for the different modeling approaches. I think the text could be improved to highlight why these are relevant metrics to use.

Lines 386-410: This paragraph might fit better into the discussion than in the results.

Reviewer #2:

Remarks to the Author:

The authors have made a thorough revision and fully addressed my comments and (though I just skimmed) I think did well for the other reviewers too. The new Fig 1, S2.3 and S2.5 are really helpful and help address my questions. Thanks for responding thoroughly to my questions about connectivity and the species whose ranges are below 60% and above 90%. The precision in the change of PGD for differentiation is also good. The additional notes about effective size in small populations are appreciated.

I look forward to seeing this novel and groundbreaking research published.

Reviewer #3:

Remarks to the Author:

The authors have now significantly improved their manuscript and the rationale for the division of life zones into proxies of genetic differentiation (PGD) is now more clearly explained.

Given that the main focus of this manuscript is on the presentation of the approach using proxies of genetic differentiation (PGD), the discussion should focus more on the strengths and weakness of this approach versus alternative ones. Using the SDMs as a baseline reference is ok, but the alternative approaches that already seek to capture infraspecific genetic diversity should be discussed and compared. This would include those simply based on geographic distance (sampling at a minimum distance) and those that also consider environmental variation (i.e. assuming that genetic differentiation is the result of divergent selection through local adaptation). In this sense, will the addition of criteria of historic processes add a better representation of genetic differentiation? Which type of genetic differentiation are we interested in identifying? Overall genetic differentiation or genetic differentiation of adaptive value? The proposed methodology considers both environmental (Holdridge Life Zones) and historical processes. How do we know whether the selected definition for both criteria is the right one? Should the territory be partitioned into a higher or a lower number of zones (with respect to Holdridge Life Zones) to capture the existing genetic differentiation? Should the division of phylogeographic patterns have been performed at a finer scale?

Specific remarks:

L 165. PGD is mentioned in this line, but it is defined in line 170. This should be corrected.

L274-284. The manuscript is complex, it has a lot of information. Therefore, it would benefit from simplifying and deleting the information that is superfluous. In my opinion, the comparison made here among the different scenarios is expendable because it is tautological that the SDM*PGD scenario is going to give the best result if the way of measuring the result is the representation of PGD within the area of each taxon.

L345-346. Authors remit to Figure 6 to show that CWR taxa may be represented at least once with a relatively little area. I don't understand how Figure 6 can show that, given that the variable in the Y axis is "mean proportion of the area of each PGD within each taxon". At the same time, the legend of this figure presents it as "Performance curves quantifying the proportion of crop wild relatives based on the hierarchical landscape priority rank map". Please, explain this better.

Table S3.1. I believe that the use of the term "crop" throughout this table to divide the life zones into PGD leads to confusion when the focus of the manuscript is on crop wild relatives. I suggest substituting it with a synonym (e.g., batch or another).

Figure 3d. I don't understand how the proportion of the area of each PGD is obtained for the two different SCP scenarios considering 20% of Mexico's terrestrial area. Is it through the use of Zonation software? If so, it should be indicated in the methods section of Supplementary Information 2 corresponding to Proxies of genetic differentiation.

Supplementary Information 1.

Methods: I don't know the specific instructions regarding the preparation of supplementary material, but it is quite tortuous for the reader to be sent from the text of the methods to figures and tables presented in the main text and in the Supplementary Information 2 and the Supplementary Information 3 documents.

Supplementary Information 2

Fig. S2.7. There is no correspondence between the text of the legend (it refers to a), b) and c)) and the number of scenarios indicated in the map (1, 2 and 3).

Incorporating evolutionary and threat processes into crop wild relatives conservation
(tracking number: NCOMMS-21-17838A).

Reply letter to Reviewers comments

=====
Dear Alicia,

Thank you again for submitting your manuscript "Incorporating evolutionary and threat processes into crop wild relatives conservation" to Nature Communications. We have now received reports from 3 reviewers and, on the basis of their comments, we have decided to invite a revision of your work for further consideration in our journal. Your revision should address all the points raised by our reviewers (see their reports below). In particular, you need to improve the clarity of the manuscript so it is accessible to a more general readership and also discuss the strengths and weaknesses of your approach as Reviewer 3 suggests.

When resubmitting, you must provide a point-by-point response to the reviewers' comments. Please show all changes in the manuscript text file with track changes or colour highlighting. If you are unable to address specific reviewer requests or find any points invalid, please explain why in the point-by-point response.

We appreciate the suggestions given by the reviewers in our previous submission. We have carried out the corrections they suggested, which we believe have improved the manuscript, specially in key aspects of its readability. Namely: we extensively modified Fig. 1 and added Extended Data Figure 1 to explain with a graphical example how the proxies of genetic differentiation were made, how they were used, and how to interpret them in analyses and figures the reviewers found hard to understand (see replies 1, 3, 4, 11, 14, 21 and 24). We also improved the way in which Fig. 3 is explained and discussed (see replies 9 -13). As for the supporting information, which indeed is extensive and was hard to follow, we merged the text and figures of the extended results and methods into a single pdf file that is considerably easier to follow (see reply 25). Very long tables are now provided as separate data files. Lastly, we extended the discussion on the strengths and weaknesses of our approach, as requested by Reviewer 3 (see replies 17-19).

REVIEWER COMMENTS

Reviewer #1 (Remarks to the Author):

The authors have revised the manuscript and resolved many of my initial comments and concerns. I have noted several parts of the paper that could still use improvement, including some improvements to the figures.

I still find the percentages of proxies of genetic differentiation to have a non-intuitive and somewhat confusing definition in this paper, and I don't fully understand how the definition matches up to a percentage. Is it that across the entire range, if a PGD is 75%, that 75% of the individuals will fall into one genetic grouping and 25% into another? Or is it that 75% of the distribution area is one cluster associated with that PGD? Do proxies overlap geographically, and how is this dealt with when it comes to these proportions/percentages? I believe this needs to be much more clearly articulated for the paper to be understood by a general audience.

Reply 1: Thank you for pointing out this issue. We now provide Extended Data Fig. 1 in order to assist the interpretation of the percentages of proxies of genetic differentiation. We also modified Figure 1 to better explain how the proxies were created.

As we have indicated, proxies of genetic differentiation were delimited using Holdridge's life zone classification system to delineate climatically distinctive zones, which were then subdivided into areas that could be potentially isolated due to biogeographic processes. Thus, each proxy has a particular distribution area, unique in shape, size, and location, that does not overlap with other proxies. The areas delimited in this way (i.e. proxies) were then overlaid with the distribution models of each taxon (i.e. proxy by taxa); thus, proxies for different CWR can only overlap when the taxon ranges overlap. We contend that the resulting areas potentially represent genetically distinctive populations, assuming that they have adapted to particular climatic conditions or have been split by historical processes. In other words, the distribution of a taxon can fall within one and up to a maximum of 102 proxies. We tested this hypothesis using existing empirical genetic data for *Zea mays* subsp. *parviglumis*.

Using these layers (i.e. proxies and proxies by taxa), we ran different scenarios of systematic conservation planning based on a different set of variables or criteria, and assessed their effectiveness by measuring the proportion of each proxy by taxa (averaged by each taxon is shown in Extended Data Fig. 1) represented within the conservation area networks of each scenario. If we were to consider the entire country, 100% of all proxies by taxa would be represented in this area. However, when only considering 20% of the national surface (based on Aichi Target 11), performance curves show that each scenario has a different effectiveness in representing the mean proportion of the area of proxies by taxa covered by the

conservation area network (represented by each point in the graph of Figure 4a and 4b). Reporting single occurrence representation in conservation planning can be a limited representation measure, therefore we report percentages as stated above to measure the effectiveness of the conservation area network and for the assessment of scenarios. We now explain this in lines 291-301.

Specific comments

Line 165 references PGD before the abbreviation is defined (defined on line 170). My personal recommendation is to not abbreviate 'proxies of genetic differentiation' at all.

Reply 2: As suggested by the reviewer, we now use the expanded name to refer to 'proxies of genetic differentiation' throughout the text and the abbreviated version in the figures.

Fig. 1: I found the figure a bit difficult to follow. I wasn't sure what the colours meant, or what the papers and lines on the 'phylogeographic patterns' section were supposed to show. I also did not understand how the puzzle pieces at the bottom fit into the overall pattern. The formatting with bolded boxes and colour in the example box drew my eye away from what the actual final output is, which is the 'Systematic conservation planning' box. I might suggest removing the example, as I found it distracting from the main point of Fig 1 (an overview of the approach) and it's somewhat repetitive with Fig 3.

Reply 3: Thanks for the observation, we re-make Fig. 1 to explain how the proxies of genetic differentiation were made, and how they are inserted as part of the 'Systematic conservation planning', which is the focus of our study. Also, we now only use black lines to connect the boxes. As stated in the legend, the bold boxes indicate expert participation and assessment, a particularly important aspect of the overall workflow to guarantee scientifically sound and relevant information to assess and implement conservation plans and programs. We eliminated the puzzle chart as requested.

Lines 186 - 188: A brief mention of what data were used (occurrences + habitats is what I understood from the methods) would improve readability of this section. Also, this was not clear from Fig 1, so consider whether the flow diagrams could be improved.

Reply 4: Thank you for highlighting this was not clear. In the new version of Fig.1 and its figure legend, we now more clearly summarize the data we used. This information is also explained in the "Species distribution modeling" section of the methods. Data on occurrences and habitat were only used for the final analysis to assess conservation areas for Mesoamerican CWR.

Lines 195 - 197: I'm not sure I follow the comparison here. It sounds like the previous study used 35 taxa, and this study used > 200, so I'm not sure I follow the statement about using only 10% of the taxa, especially since the following sentence states that

167 taxa were estimated in a 15x15km grid, which sounds like more than the estimates reported here estimates. Furthermore, Fig. 2 shows the highest category for taxa as 31-52, so differentiating between the 35 taxa vs >50 is not possible in the graph. I suggest re-wording parts of this section for clarity, and possibly consider re-making Fig. 2 with narrower groupings if this comparison is important.

Reply 5: We rephrased lines 197-202 of this section and included the numbers of CWR taxa used in each analysis for more clarity and to emphasize the point that global analysis might underestimate the number of CWR taxa in Mexico. We also included the number of taxa used in each analysis in Fig. 1.

Fig. 2: I'm not sure if this figure is entirely necessary to include in the main text, especially if the authors are motivated to reduce the number of main text colour figures.

Reply 6: Species richness maps allow the reader to compare the richness pattern of CWR to the results of the systematic conservation planning analysis. Thus, it can be noted that the proposed conservation area does not only focus on sites with high taxa richness, but also on low taxa richness areas where limited-range taxa might occur, thus optimizing the representation of all CWR taxa within PGD in the proposed conservation network.

As we do not exceed the number of displayed items in the ms, and the editor did not ask us to reduce the number of figures, we would like to keep Fig. 2.

Lines 212-214: Consider signalling here that the proxies mirror genetic differentiation in at least your 1 example

Reply 7: Done. As the example is explained later in the text, we did not specifically describe the example in this sentence.

Lines 234-235: The phrasing of this sentence was a bit confusing. Consider instead something along the lines of "Zea mays ... was not included in the literature review so as not to have its phylogeographic patterns influence the estimation of proxies for genetic diversity for this species. This approach allowed us to mimic the scenario presented by the many species without any existing genomic resources."

Reply 8: Done, thank you for your suggestion.

Lines 237 - 241 and Fig. 3: The figure and the text here seem to show that there is some isolation by distance, which makes the inferences of clusters difficult to determine. How much does this impact your interpretations?

Reply 9: There is indeed isolation by distance in *Z. mays* subsp. *parviglumis*. This is mostly given by populations that fell in PGD 37 and 43, where a longitudinal pattern of differentiation can be seen (Fig. 3a). However, there are other highly structured populations (e.g. represented by PGD 41 or 11), which we now highlight in the main text (line 255). Normally, systematic conservation planning would use SDM without differentiating genetic clusters within the taxon distribution range, thus stochastically representing genetic groups based on their geographic extent. For example,

populations that fell in PGD 41 (bright green admixture group in Fig. 3a) and 11 (orange admixture group in Fig. 3a), are less represented in the Zonation solution using only SDM (Fig. 3d).

Additionally, we now discuss (see Replies 17 and 18) that previous studies comparing surrogates of genetic diversity to genetic data have found that although isolation by distance is a common pattern, distance alone tends to not be a good surrogate for representing broad-scale genetic diversity, because it tends to miss genetically distinct groups of populations (Hanson 2021), which we now mention in lines 221-224 and in Fig. 3 figure legend.

Hanson, *et al.* 2021. "Evaluating Surrogates of Genetic Diversity for Conservation Planning". *Conservation Biology* 35 (2): 634–42. <https://doi.org/10.1111/cobi.13602>.

Lines 241 - 246: This section was very confusing to me as written. In what way did the approach increase representation of areas? Maybe point to the specific part of the figure that shows this (Fig. 3d, I think). I was unclear what "there is no complete coincidence between the PGD and the empirical genetic data" actually means, or where this is shown.

Reply 10: We rewrote this part (lines 259-264, 269-270, 279-282) to help the reader better follow Fig. 3, and added the following example: the Zonation scenario using only SDM favors the representation of populations that fell in the proxies 37, 5, 41 and 36 (Fig. 3d) and poorly represents populations that fell in proxies 8 and 40 (dark blue admixture group, Fig. 3a). Alternatively, the Zonation scenario dividing SDM by proxies of genetic differentiation, increases the representation of all proxies and no admixture group is poorly represented (Fig. 3d).

Fig. 3: I found several parts of this figure to be difficult to follow. 3a and 3b are reasonably straightforward, although some of the colours are difficult to tell apart (the two yellows in the admixture plot, the two brownish greens in the PGD topologies). But the points were very small and the colours almost impossible to tell apart in Fig 3c, and the legend should specify that this was a PCA of the genetic data. For 3d, I was confused about what the proportion is and how it can be assigned to each proxy of genetic differentiation, especially in models without the proxies for genetic differentiation. Should the proportions sum to 1?

Reply 11: As for Fig. 3c, after trying different point sizes, we found this one to be the most adequate because larger point sizes overlap, hiding some of the proxies entirely. Thanks for noticing the figure legend did not specify the PCA was of genetic data; this has been corrected.

We understand Fig. 3d is not straightforward to interpret, but we believe it would be easier to follow now that we added an Extended Data Fig. 1 to show how the mean proportions were estimated and what they mean. See also Reply 1. We also added an example in the main text (see reply 10) using Fig. 3d.

To me, it seems like the main goal of Fig. 3/the analysis with *Zea mays* is to demonstrate that the proxies of genetic differentiation are decent proxies, and I'm not entirely convinced.

Reply 12: The purpose of Fig. 3 is to show that the proxies of genetic differentiation, although not perfect, are better at representing the genetic variation in the Zonation output than if using SDM alone. We believe this proof of concept is worth sharing with the systematic conservation planning community. Of course, the more refined the proxies are (on an extreme, they could even be based on genetic data) the better the actual genetic structure would be represented. We now briefly comment on this in lines 285-287. See also replies 9-11 and 13.

A secondary goal of Fig. 3 is to demonstrate that the SDM + PGD is better than just the SDM, but I think Fig. 4 does a better job of showing this, so I would suggest removing 3d from Fig 3 as it detracts from the primary goal of the figure.

Reply 13: We would like to keep Fig. 3d because it was asked by Reviewer 2 in the previous round and also because we believe Fig. 3d helps to understand Fig. 4, as it represents what is behind each data point of Fig. 4; i.e. the average of all the grey bars represents the data point of *Zea mays* subsp. *parviglumis* under the SDM scenario (orange) of Fig. 4. This is now also explained in its legend.

Fig 4 and lines 260 - 275: I find the use of the proportion of taxa distributions and proportion of PGD areas confusing metrics of performance for the different modeling approaches. I think the text could be improved to highlight why these are relevant metrics to use.

Reply 14: Thanks for the observation. We now included an Extended Data Fig. 1 to assist interpreting the percentages of proxies of genetic differentiation. Please see reply 1.

Lines 386-410: This paragraph might fit better into the discussion than in the results.

Reply 15: Our MS follows the Author guidelines of the journal. Accordingly, the results and discussion are combined in the Results section, while the last section summarizes the main conclusions in a few short paragraphs.

Reviewer #2 (Remarks to the Author):

The authors have made a thorough revision and fully addressed my comments and (though I just skimmed) I think did well for the other reviewers too. The new Fig 1, S2.3 and S2.5 are really helpful and help address my questions. Thanks for responding thoroughly to my questions about connectivity and the species whose ranges are below 60% and above 90%. The precision in the change of PGD for differentiation is also good. The additional notes about effective size in small populations are appreciated.

I look forward to seeing this novel and groundbreaking research published.

Reply 16: Thank you very much for your comments in the first round that helped us to improve the ms and for your positive feedback on the changes we made.

Reviewer #3 (Remarks to the Author):

The authors have now significantly improved their manuscript and the rationale for the division of life zones into proxies of genetic differentiation (PGD) is now more clearly explained.

Given that the main focus of this manuscript is on the presentation of the approach using proxies of genetic differentiation (PGD), the discussion should focus more on the strengths and weakness of this approach versus alternative ones. Using the SDMs as a baseline reference is ok, but the alternative approaches that already seek to capture infraspecific genetic diversity should be discussed and compared. This would include those simply based on geographic distance (sampling at a minimum distance) and those that also consider environmental variation (i.e. assuming that genetic differentiation is the result of divergent selection through local adaptation). In this sense, will the addition of criteria of historic processes add a better representation of genetic differentiation?

Reply 17: We have extended the section “Overcoming lack of genetic data with spatial surrogates” to compare our approach with previous approaches seeking to capture infraspecific genetic diversity, as well as to better justify why historical processes leading to population structure are relevant for conservation and use of genetic diversity (lines 216-229). Shortly, population structure can result in locally restricted alleles, both neutral and of adaptive value, and thus population structure should be accounted for when aiming to represent genetic diversity. Previous studies focusing only on environmental variables or distance have shown to be incomplete surrogates, so we propose adding historical criteria to complement them.

Regarding the weaknesses and strengths of our approach, the main caveat of our methodology is that the spatial expression of the proxies of genetic differentiation would never be as accurate as actual genetic studies, which we now state in lines 279-282 and contrast it with the main strengths 285-287. The lack of genetic data to corroborate if the proxies are reliable was already a limitation, which was already mentioned in lines 429-431. We also now briefly mention in the discussion that as better phylogeographic metanalyses and genetic data become more available, proxies of genetic differentiation could be fine-tuned or described at higher resolutions (lines 473-475).

Which type of genetic differentiation are we interested in identifying? Overall genetic differentiation or genetic differentiation of adaptive value?

Reply 18: Our study focuses on crop wild relatives, so genetic diversity of adaptive value is of special interest due to its potential usefulness in plant breeding. However, genetic differentiation can lead to speciation by purely neutral processes, and therefore it is of conservation interest even if such differentiation is not of adaptive value (see reply 17 and new lines 216-229). Our methodology can therefore be useful for conservation planning in general.

The proposed methodology considers both environmental (Holdridge Life Zones) and historical processes. How do we know whether the selected definition for both criteria is the right one? Should the territory be partitioned into a higher or a lower number of zones (with respect to Holdridge Life Zones) to capture the existing genetic differentiation? Should the division of phylogeographic patterns have been performed at a finer scale?

Reply 19: Thank you for raising these questions, which we would like to address in follow-up studies. But to briefly answer your concerns:

Our rationale is based on the assumption that environmental differences are a first proxy of genetic differentiation (that has already been used by other studies) because different environmental conditions should lead to differentiation at adaptive loci and local adaptation. To define environmental variation spatially, we used Holdridge life zones because they summarize continuous climate data of several variables (which are known to specially affect plants, see Supplementary Table 1.8) into a single layer. Also, Holdridge life zones are based on data available to any part of the world at different resolutions (1 km² for Mexico; Supplementary Fig. 1.2), thus making our method easier to apply for other regions. Our method, however, could be applied to other data (for instance, soil type) or fine-tuned to climate variables of particular interest.

As for accounting for historical processes, we proposed subdividing the life zones and then using this information to subdivide the SDM because this approach allows subdividing each taxon range following a single reproducible criterion and without *a priori* genetic data (Supplementary Table 1.9). A possible alternative to our approach would be to subdivide each SDM according to idiosyncratic reasons for each taxa. For instance, based on climate variables that have proved to influence adaptive traits, and on prior phylogeographic studies for that taxon. However, for this to be possible, genetic studies with large sampling would be necessary, which is not the case for our study taxa, but may be possible for other groups.

Regarding the scale and resolution of the PGDs, here we targeted phylogeographic patterns to represent general trends that would likely hold across species, because it is a new approach, and we wanted to be as conservative as possible. As more information on phylogenetic patterns becomes available, life zones could be further subdivided. Alternatively SDM subdivision by PGDs could be fine-tuned based on genetic studies on the target taxa, and be incorporated into the spatial assessment of Zonation.

Still, there might be arguments to merge two or more life zones or PGDs into one single unit if there is no evidence of genetic differentiation between different

groups. However, as our aim is to conserve genetic diversity, following a precautionary principle, it is better to assume genetically distinct areas as not to do so.

Specific remarks:

L 165. PGD is mentioned in this line, but it is defined in line 170. This should be corrected.

Reply 20: Thanks for the observation. As suggested by reviewer 1, we now use the expanded name to refer to 'proxies of genetic differentiation' throughout the text and the abbreviated version in the figures (please see Reply 2).

L274-284. The manuscript is complex, it has a lot of information. Therefore, it would benefit from simplifying and deleting the information that is superfluous. In my opinion, the comparison made here among the different scenarios is expendable because it is tautological that the SDM*PGD scenario is going to give the best result if the way of measuring the result is the representation of PGD within the area of each taxon.

Reply 21: We agree the manuscript is complex, thus, we keep making an effort to simplify the information as much as possible and follow the reviewer's comments, eg., we integrated the three Supplementary Materials into one single document (please see reply 25).

Regarding Fig. 4, we have acknowledged that this measurement is somewhat redundant (lines 299-301), as we have no other way to measure genetic diversity in a spatially explicit matter. Nonetheless, we believe it is critical to show in a quantitative way that less complex options to include proxies of genetic differentiation in the conservation assessment (eg. as individual conservation features, scenario (iii) SDM+PGD; or administrative planning units, scenario (v) SDM and PGD as administrative units) offer suboptimal (less efficient) results for the representation of potential genetic variation of a taxon in the solution. We believe that Fig. 4 is an important figure in systematic conservation planning papers. In order to support its interpretation, we provided an Extended Data figure. Please see reply 1 and changes in lines 291-301.

L345-346. Authors remit to Figure 6 to show that CWR taxa may be represented at least once with a relatively little area. I don't understand how Figure 6 can show that, given that the variable in the Y axis is "mean proportion of the area of each PGD within each taxon". At the same time, the legend of this figure presents it as "Performance curves quantifying the proportion of crop wild relatives based on the hierarchical landscape priority rank map". Please, explain this better.

Reply 22: You are right, Fig. 6 does not explicitly show this result; we analyzed the raw data that allowed us to make this statement. We changed the wording to be more precise and only refer to what Fig. 6 shows (lines 386-387).

Regarding the figure legend, the idea is that the reader can associate the taxa representation curves, in particular the area percentages of the y-axis, to the hierarchical landscape priority rank map. We adjusted the legend to be clearer. We also better explain how to interpret the hierarchical landscape map in lines 441-444.

Table S3.1. I believe that the use of the term “crop” throughout this table to divide the life zones into PGD leads to confusion when the focus of the manuscript is on crop wild relatives. I suggest substituting it with a synonym (e.g., batch or another).

Reply 23: Thank you for this suggestion, we replaced “crop” for “division”.

Figure 3d. I don't understand how the proportion of the area of each PGD is obtained for the two different SCP scenarios considering 20% of Mexico's terrestrial area. Is it through the use of Zonation software? If so, it should be indicated in the methods section of Supplementary Information 2 corresponding to Proxies of genetic differentiation.

Reply 24: To answer this concern we added an Extended Data Fig. 1. See also replies 1 and 11.

Supplementary Information 1.

Methods: I don't know the specific instructions regarding the preparation of supplementary material, but it is quite tortuous for the reader to be sent from the text of the methods to figures and tables presented in the main text and in the Supplementary Information 2 and the Supplementary Information 3 documents.

Reply 25: We now integrated the three documents into one single document, by describing the methods, followed by the results of each step of the working process. As the idea is to provide a document that the reader can easily follow, we now provide large tables as additional excel files.

Supplementary Information 2

Fig. S2.7. There is no correspondence between the text of the legend (it refers to a), b) and c)) and the number of scenarios indicated in the map (1, 2 and 3).

Reply 26: The figure, now Supplementary Fig. 1.9, shows the coincidence of the proposed conservation scenarios a, b, and c; thus, the legend indicates *how many* scenarios concur: 1- only one of the three scenarios is represented, i.e. no coincidence of scenarios 2- the combination of two scenarios is shown, i.e. scenarios a and b, scenarios a and c, or scenarios b and c; 3- all three scenarios coincide. In neither case, it is indicated which scenarios. The geographical areas with high conservation value for Mesoamerican CWR in Mexico can be found in Supplementary Figs. 1.12 and 1.13.

Reviewers' Comments:

Reviewer #1:

Remarks to the Author:

I thank the authors for their thoughtful responses to my previous review. My major concerns have been addressed, and I only have a few minor comments to improve understanding a flow in a few parts of the manuscript.

Lines 150-171: When introducing the study, I suggest that the authors explicitly mention that they are focused on identifying ways to improve conservation outcomes through managing 20% of Mexico's landmass – and perhaps provide a justification for this figure. It pops up throughout the methods/results, but is not clearly introduced as a goal so can be a bit confusing to the reader. For example, move the sentence on lines 343-344 to the introduction/early results section.

Line 194: Rephrase the second half of the sentence ('which are known for...') for improved grammar/flow, for example to 'regions which are known for harbouring taxa with high genetic differentiation'.

Line 223: Replace 'it tends to' to 'has potential to'.

Lines 225-228: change 'include' and 'define' to 'including' and 'defining' to match the phrasing of 'we focused on'.

Line 227: change 'them' to say 'proxies' to improve clarity.

Lines 245-247: In my opinion, this part of the text should make it clear that proxies of genetic differentiation are areas, similar to range maps, not genetic differentiation metrics (e.g., proxies of F_{st}). The authors gave a very clear description of this in their response to reviewers, and I think that a similar explanation should occur in the main text – I would not say that the lines on 291-301 successfully accomplish this task.

Line 248: I found the first sentence in this paragraph to be a bit confusing. I suggest rewriting it to say something more like, "To test how well our method for identifying proxies of genetic differentiation worked, we leveraged an available empirical dataset for the teosinite..." I think this will more clearly communicate both what was done and why the authors focus on this species in particular in this section.

Lines 291-311: This paragraph could use a more concise topic sentence. I think this paragraph is referring to extending the modeling approach tested in the maize relative to all of the other taxa of interest, but it's not clear from the text. Adding a clear topic sentence would help substantially.

Lines 304-307 and Extended Data Fig. 1: I'm very confused by these numbers and how the text relates to the figure. Why is only 20% of the country being evaluated? Where do 41% of the taxon range and 76% of the area fit into the graph? What do the two scenarios in (a) and (b) in the figure represent? Are the proposed conservation areas the same region in (a) and (b)? From the figure caption, I can tell that (a) and (b) are meant to be two different scenarios, but I'm confused by whether the different ranges/proxies are meant to represent different species or different outcomes of the species distribution models (e.g., with and without proxies included). I think most of this confusion could be resolved by revising the figure caption, but I also suggest the authors try to clarify within the main text as well (i.e., be more specific in the sentence, beyond 'Based on this scenario', which is vague – I'm not sure what scenario is being referred to).

Line 375: Replace 'Noteworthy' with 'Of note' to improve grammar.

Reviewer #3:

Remarks to the Author:

The authors have addressed successfully all the concerns and suggestions that I raised. I also

believe they have addressed properly the suggestions of the other two reviewers. I congratulate the authors for the efforts carried out to improve their manuscript.

Just two minor points:

Legend of Extended Data Fig. 1. "(a) All proxies, and consequently the full taxon range, is..." substitute "is" by "are".

Lines 386-389: Although the authors mention in the rebuttal letter that they have addressed my concern, I still think "Fig.6" should not be placed after "Although CWR taxa might be represented within a relatively small area (i.e. in less than 2% of Mexico)", because Fig. 6 does not illustrate this. I would suggest it is placed at the end of the sentence, because Fig 6 effectively shows that "the conservation of ecological and evolutionary processes shaping biodiversity at all levels...can not be secured in a small fraction of the territory and with few individuals".

Reply letter

We would like to thank the reviewers for the positive feedback on the previous versions of the manuscript. Also, we are thankful for the very detailed review and helpful comments to improve this last version.

REVIEWERS' COMMENTS

Reviewer #1 (Remarks to the Author):

I thank the authors for their thoughtful responses to my previous review. My major concerns have been addressed, and I only have a few minor comments to improve understanding a flow in a few parts of the manuscript.

Lines 150-171: When introducing the study, I suggest that the authors explicitly mention that they are focused on identifying ways to improve conservation outcomes through managing 20% of Mexico's landmass – and perhaps provide a justification for this figure. It pops up throughout the methods/results, but is not clearly introduced as a goal so can be a bit confusing to the reader. For example, move the sentence on lines 343-344 to the introduction/early results section.

Reply 1: Thank you for pointing out this topic. The 20% area was not assessed *a priori*, it resulted from the efficiency in representing the genetic diversity of CWR taxa. Therefore, we would not like to introduce this figure in the introduction. We have rephrased lines 149-152 to clarify that the conservation assessment was conducted to enable concentrated management efforts by identifying a portion of the country that maximizes the representation of genetic diversity.

Line 194: Rephrase the second half of the sentence ('which are known for...') for improved grammar/flow, for example to 'regions which are known for harbouring taxa with high genetic differentiation'.

Reply 2: Done. Lines 190-191.

Line 223: Replace 'it tends to' to 'has potential to'.

Reply 3: Done.

Lines 225-228: change 'include' and 'define' to 'including' and 'defining' to match the phrasing of 'we focused on'.

Reply 4: Done.

Line 227: change 'them' to say 'proxies' to improve clarity.

Reply 5: Done.

Lines 245-247: In my opinion, this part of the text should make it clear that proxies of genetic differentiation are areas, similar to range maps, not genetic differentiation metrics (e.g., proxies of F_{st}). The authors gave a very clear description of this in their response to reviewers, and I think that a similar explanation should occur in the main text – I would not say that the lines on 291-301 successfully accomplish this task.

Reply 6: Done. As suggested, we have included a short explanation in the section 'Overcoming lack of genetic data with spatial surrogates' (lines 245-247), and also provided some more details in the section 'Conservation areas for Mesoamerican CWR in Mexico' (lines 336-338).

Line 248: I found the first sentence in this paragraph to be a bit confusing. I suggest rewriting it to say something more like, "To test how well our method for identifying proxies of genetic differentiation worked, we leveraged an available empirical dataset for the teosinte..." I think this will more clearly communicate both what was done and why the authors focus on this species in particular in this section.

Reply 7: Done. We rephrased as suggested. Lines 249-250.

Lines 291-311: This paragraph could use a more concise topic sentence. I think this paragraph is referring to extending the modeling approach tested in the maize relative to all of the other taxa of interest, but it's not clear from the text. Adding a clear topic sentence would help substantially.

Reply 8: Done. We rephrased the first part of the paragraph to be clearer (lines 334-335). For the final analysis, we used the modeling approach that best captured the potential genetic variation inferred through proxies, i.e. CWR potential distribution models subdivided by proxies of genetic differentiation. This analysis was different and independent from the analysis to evaluate the proxies of genetic differentiation for which we used the empirical data from the wild relative of maize.

Lines 304-307 and Extended Data Fig. 1: I'm very confused by these numbers and how the text relates to the figure. Why is only 20% of the country being evaluated? Where do 41% of the taxon range and 76% of the area fit into the graph? What do the two scenarios in (a) and (b) in the figure represent? Are the proposed conservation areas the same region in (a) and (b)? From the figure caption, I can tell that (a) and (b) are meant to be two different scenarios, but I'm confused by whether the different ranges/proxies are meant to represent different species or different outcomes of the species distribution models (e.g., with and without proxies included). I think most of this confusion could be resolved by revising the figure caption, but I also suggest the authors try to clarify within the main text as well (i.e., be more specific in the sentence, beyond 'Based on this scenario', which is vague – I'm not sure what scenario is being referred to).

Reply 9: We adjusted the legend and panels of the Extended Data Fig. 1 (now Supplementary Figure 7) to match the numbers of the illustrative example in the figure (Panel b) with those in the main text. Panel (a) provides a comparative example that shows the full coverage of a given taxon range and its proxies.

We choose 20% of the country based on the performance curves to efficiently represent taxa ranges delimited by proxies (Fig. 6), and also provide a reference to Aichi Target 11, which recommends to protect at least 17% of terrestrial areas (Fig. 6).

Line 375: Replace 'Noteworthy' with 'Of note' to improve grammar.

Reply 10: Done.

Reviewer #3 (Remarks to the Author):

The authors have addressed successfully all the concerns and suggestions that I raised. I also believe they have addressed properly the suggestions of the other two reviewers. I congratulate the authors for the efforts carried out to improve their manuscript.

Just two minor points:

Legend of Extended Data Fig. 1. "(a) All proxies, and consequently the full taxon range, is..." substitute "is" by "are".

Reply 11: Done.

Lines 386-389: Although the authors mention in the rebuttal letter that they have addressed my concern, I still think "Fig.6" should not be placed after "Although CWR taxa might be represented within a relatively small area (i.e. in less than 2% of Mexico)", because Fig. 6 does not illustrate this. I would suggest it is placed at the end of the sentence, because Fig 6 effectively shows that "the conservation of ecological and evolutionary processes shaping biodiversity at all levels...can not be secured in a small fraction of the territory and with few individuals".

Reply 12: Done.